# Ocean warming drives immediate mass loss from calving glaciers in the high Arctic

Ø. Foss ⓘ[1] ✉, J. Maton[1], G. Moholdt ⓘ[1], L. S. Schmidt ⓘ[2], D. A. Sutherland . ⓘ[3], I. Fer ⓘ[4,5], F. Nilsen[6], J. Kohler[1] & A. Sundfjord ⓘ[1]

Glaciers in the Arctic have lost considerable mass during the last two decades. About a third of the glaciers by area drains into the ocean, yet the mechanisms and drivers governing mass loss at glacier calving fronts are poorly constrained in part due to few long-term glacier-ocean observations. Here, we combine a detailed satellite-based record of calving front ablation for Austfonna, the largest ice cap on Svalbard, with in-situ ocean records from an offshore mooring and modelled freshwater runoff for the period 2018-2022. We show that submarine melting and calving occur almost exclusively in autumn for all types of outlet glaciers, even for the surging and fast-flowing glacier Storisstraumen. Ocean temperature controls the observed frontal ablation, whereas subglacial runoff of surface meltwater appears to have little direct impact on the total ablation. The seasonal warming of the offshore waters varies both in magnitude, depth and timing, suggesting a complex interplay between inflowing Atlantic-influenced water at depth and seasonally warmed surface water in the Barents Sea. The immediate response of frontal ablation to seasonal ocean warming suggests that marine-terminating glaciers in high Arctic regions exposed to Atlantification are prone to rapid changes that should be accounted for in future glacier projections.

Mass loss from glaciers in the Arctic, including Svalbard has in the last two decades been comparable to the mass loss from the major ice sheets of Antarctica and Greenland[1]. During the same period, the Svalbard region has undergone significant environmental changes, with strongly increasing air temperatures[2] and reduced sea ice cover[3], as well as warming surface and subsurface ocean temperatures offshore in the northwestern Barents Sea[4,5]. The majority of glaciers in eastern Svalbard drain into the ocean, and as much as a quarter of the Austfonna ice cap is grounded below sea level[6], making this a region where glaciers are highly susceptible to ocean influence[7]. Yet, it is unclear to what degree ocean forcing is a driver of ongoing glacier mass loss in the area.

Long-term mass loss from marine-terminating glaciers is strongly influenced by changes in atmospheric and oceanic temperatures[8], which are in turn governed by slowly changing patterns of, e.g., winds and ocean circulation. However, the seasonal environmental variability may be much greater than the interannual and can therefore yield important insights into how glaciers respond to environmental forcing[9]. Seasonal variations in frontal ablation rates typically result from a combination of changes in the ocean water column at the glacier front and subglacial freshwater runoff, both of which exhibit strong seasonality (e.g[9,10]). Frontal ablation is the sum of calving and melting, the latter being strongly influenced by buoyant meltwater plumes that rise along the glacier terminus, drawing in ambient ocean water and transferring heat from the plume to the ice [11–13]. Consequently, frontal melt generally increases with both ocean temperature and freshwater flux. The relative importance of these two driving factors—one atmospherically driven and the other oceanic—can vary significantly across different systems.

[1]Norwegian Polar Institute, Tromsø, Norway. [2]University of Oslo, Oslo, Norway. [3]University of Oregon, Eugene, OR, USA. [4]Geophysical Institute, University of Bergen, Bergen, Norway. [5]Bjerknes Center for Climate Research, Bergen, Norway. [6]The University Centre in Svalbard, Longyearbyen, Norway. ✉e-mail: oyvind.foss@npolar.no

In this study, we investigate the seasonal evolution of marine-terminating glaciers on Austfonna (Fig. 1) over a six-year period from autumn 2016 to autumn 2022. We analyse the evolution of frontal ablation (the sum of calving and melting at the glacier front) for all glaciers during 2016-2021, with a particular focus on the Storisstraumen glacier during 2018−2022. During this period, coincident time series of modelled meltwater runoff from the glacier and ocean temperatures from an offshore mooring are available, allowing us to evaluate the influence of different environmental drivers on frontal ablation on the Barents Sea margin of Svalbard.

## A major ice cap in a unique oceanographic setting

Austfonna ice cap, one of Europe's largest glaciers, covers 57% (8100 km²) of Nordaustlandet island. It feeds 28 drainage basins, 19 of which terminate into the sea (Fig. 1d, c), chiefly toward the east and south. The calving fronts of Austfonna have a total length of ~260 km, constituting around 28% of Svalbard's total tidewater glacier terminus

length. No persistent ice shelves or floating tongues have been observed on Svalbard[14], and glaciers on Austfonna are far from the flotation limit when comparing terminus freeboard with bathymetry (e.g. Fig. 1f).

Located between the Nansen Basin of the Arctic Ocean and the cold, seasonally ice-covered northern Barents Sea, Nordaustlandet inhabits an oceanographic environment distinct from the warmer and largely ice-free waters offshore western Svalbard (Fig. 1c). Sea ice presence inhibits ocean mixing and limits surface warming during ice-covered parts of summer. Ocean waters below the mixed layer at the northwestern margin of the Barents Sea consist of relatively cold, Arctic water masses, but are influenced by the Fram Strait branch of the Atlantic Water boundary current which flows along the continental slope north of Svalbard. The boundary current increases in temperature and volume in autumn[15,16], and branches of Atlantic-influenced waters spill over southward into the northern Barents Sea though submarine troughs during autumn and early winter[17,18], resulting in a

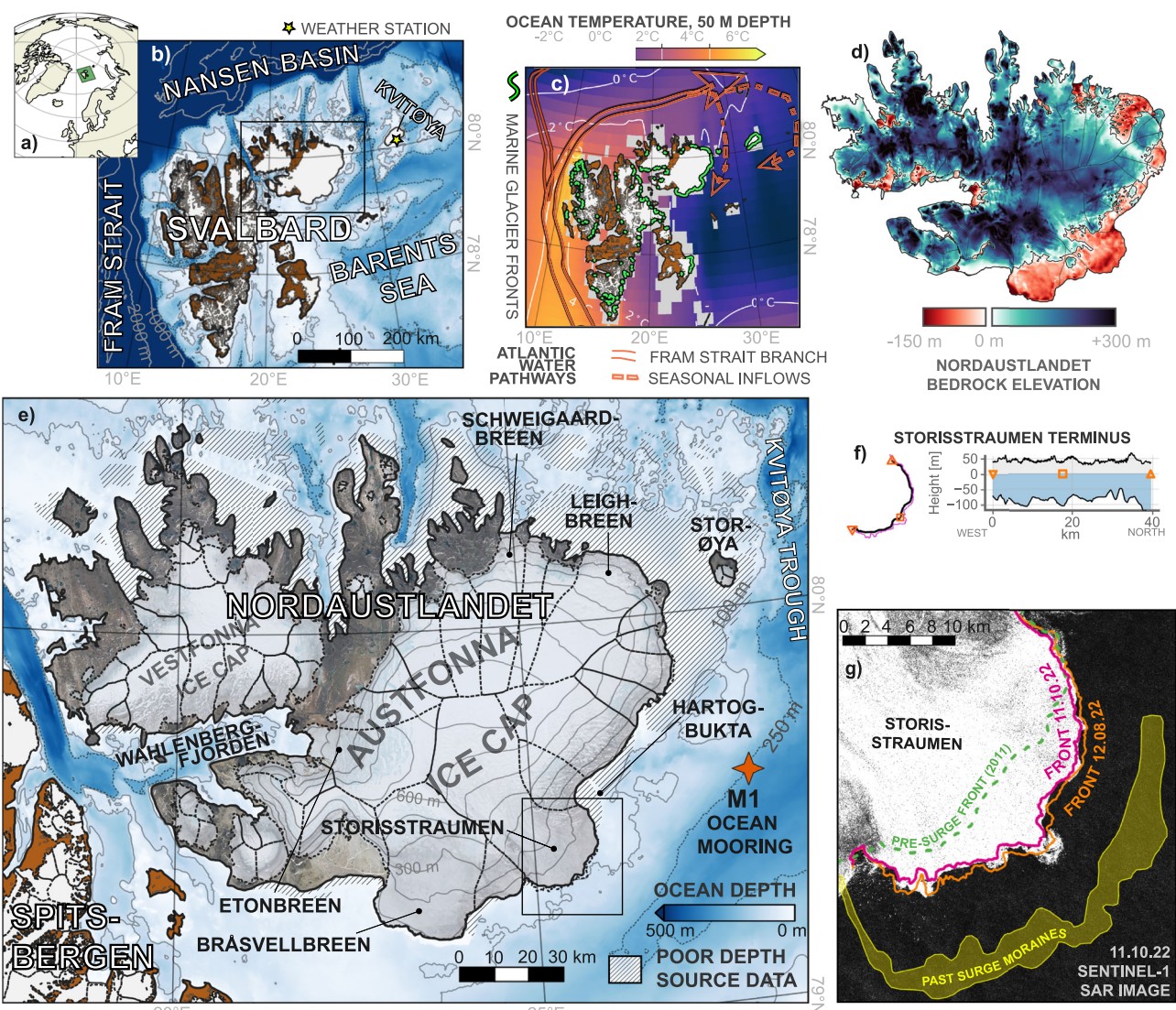

**Fig. 1 | Overview of Austfonna ice cap and Storisstraumen glacier. a** Location of Svalbard. **b** Regional map with ocean bathymetry. Yellow star shows Kvitøya weather station[67], black rectangle indicates the area shown in panel **e. c** Mean ocean temperature at 50 m depth from the NOAA Arctic Ocean Regional Climatology[68]. Orange arrows show a simplified schematic of Atlantic Water circulation pathways through Fram Strait including seasonal inflows into the northern Barents Sea[17]. Green lines show marine glacier fronts[56]. **d** Nordaustlandet bedrock elevation[60]. **e** Map of Austfonna with topography and mooring position. Hashed areas indicate areas regions with poor source bathymetry data (see Fig. S2). Black rectangle indicates the area shown in panel **g. f** Bed[60] and surface[61] elevation along the Storisstraumen terminus (length from a simplified geometry shown in black on the left). **g** Sentinel-1 SAR image of Storisstraumen, showing examples of extracted front positions, the pre-surge (2011) front, and past surge moraine region[22]. Bathymetry in panels b and e are from IBCAO v4.0[69].

relatively deep warm layer with a core around 150 to 200 m depth[19,20]. The respective seasonal patterns of surface ocean warming and deeper Atlantic Water inflow both tend to result in water temperature peaks months after the maximum in air temperature and glacier surface melt. This rather unique feature of the physical environment makes it possible to separate the effects of freshwater runoff and ocean thermal forcing in time.

## Results and Discussion

### Glacier frontal ablation coincides with seasonal ocean warming

Frontal ablation of the most active glacier on Austfonna, the surging Storisstraumen, largely occurred in autumn (Fig. 2) and generally coincided with the warming of the coastal ocean as observed at the ocean mooring located ~50 km from the glacier front (Fig. 1a). The autumn peak in ocean temperature and frontal ablation is distinct from the summer peak in surface melting and resulting subglacial freshwater discharge at the glacier terminus, implying that discharge is not a direct driver of the seasonal retreat of the glacier. The ocean-controlled seasonal pattern of frontal ablation is common for all types of marine-terminating glaciers on Svalbard, ranging from fast-flowing outlets to near-stagnant ice tongues facing the Barents Sea (Fig. 4c). This is consistent with frontal mass loss driven by thermal ocean forcing and suggests a profound influence of regional ocean circulation and heat content on the mass balance of marine-terminating glaciers in this part of the high Arctic.

A simple analysis using monthly values confirms significant positive correlations between frontal ablation and offshore ocean *thermal driving* (water temperature above freezing); $r = 0.93$ and $r = 0.42$ for ~20 m and ~95 depth, respectively (Fig. S3). The poorer correlation at 95 m depth may be a result of lateral gradients between the offshore temperatures and the water column near the glacier face. Although we will show below that deep heat transport toward the coast does occur, we do not know whether it occurs systematically or throughout all seasons, and the mixed layer depth may, for example, deepen toward the coast during parts of the year. We also observe a significant correlation ($r = 0.75$) between frontal ablation and offshore sea ice concentration—most likely because both variables are correlated with upper ocean temperature rather than sea ice directly impacting frontal ablation (especially since there is little fast ice in the area, see Fig. S1c).

No significant correlation was found between frontal ablation and modelled freshwater runoff. During parts of the year where ocean thermal forcing offshore of Storisstraumen was weak (<-1° C), there was little to no frontal ablation to balance the ice discharge from the glacier interior, and the glacier advanced into the surrounding ocean (Fig. 2a). Calving and terminus melting were minimal in the absence of ocean forcing, despite the rapid flow and widespread crevassing of the glacier throughout the year. Considering that Storisstraumen has been in a surging fast-flow mode since 2012[21], we hypothesize that the glacier would have advanced much further into the Barents Sea if ocean conditions were colder. The location of offshore moraines (Fig. 1f) shows that the glacier extended ~10 km further into the ocean during the previous surge of Storisstraumen some time before the mid-19th century[22].

Our findings are consistent with observations of seasonal, ocean-driven retreat of three tidewater glaciers in western Spitsbergen during 2013-2014[23]. However, the peak of glacier retreat at those western

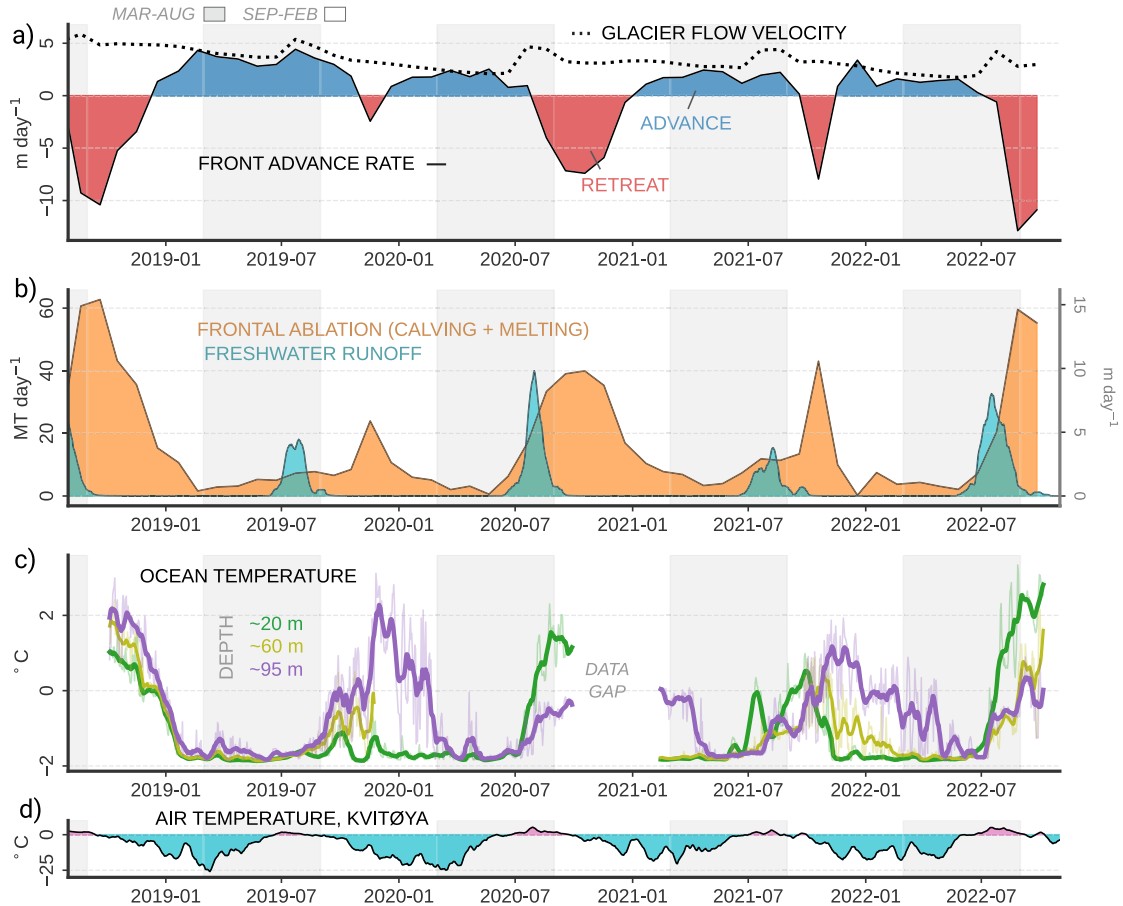

**Fig. 2 | Time series from Storisstraumen glacier and offshore ocean temperatures. a** Glacier front retreat/advance of Storisstraumen and mean near-front glacier flow velocity. **b** Storisstraumen frontal ablation and modelled freshwater runoff (14-day rolling mean). **c** Ocean temperature at the M1 mooring (thick lines: 14-day rolling mean, thin lines: daily means). **d** Air temperature from a coastal automatic weather station at Kvitøya[67] (14-day rolling mean).

glaciers occurred in August-September, several months before Storisstraumen in the present study. We attribute this difference to the earlier onset of ocean warming on the west coast. The delayed ocean warming on the eastern side of Svalbard can be explained by i) the later arrival of the annual maximum of relatively warm Atlantic Water as the seasonal pulse propagates around the archipelago[24,25], and ii) the presence of sea ice in the northern Barents Sea in spring into summer[17,26]. The latter delays surface warming or restricts it to a thin surface meltwater layer, compared to the relatively warm and largely sea ice-free waters offshore of western Spitsbergen.

The relatively late-season onset of ocean warming in northeastern Svalbard clearly separates the annual ocean temperature peak from the peak in meltwater runoff that is largely controlled by atmospheric temperature. The observed time lag of 1-3 months between the July-August peak in modelled runoff and the autumn peak in glacier retreat is greater than the bulk time scale of meltwater transport from the glacier surface to the grounding line expected for subglacial runoff. This is especially the case since surface melt is most intense at lower elevations, closer to the calving fronts and in more crevassed areas[27]. During all measurement years, the runoff peaks coincided with peaks in the glacier flow velocity (Fig. 2a). This acceleration of the glacier flow in the runoff season indicates that the surface meltwater rapidly percolated through the glacier to the bedrock, lubricating the base of the glacier and increasing its flow velocity[28], consistent with earlier in-situ observations from GNSS stations on the glacier[21]. Observations of widespread meltwater plumes in July and August[29] imply that surface meltwater exits into the ocean as subglacial discharge relatively quickly, supporting the interpretation that driving mechanisms other than runoff must control the bulk of the frontal ablation.

Previous estimates have indicated that surface melt runoff is the largest negative term in the mass balance of Svalbard glaciers, with frontal ablation as sizeable but smaller contributor (20–50% of runoff[30], ~50% for Austfonna[31]). This study shows that frontal ablation was the dominant mass loss term at Storisstraumen between 1 September of 2018 and 2022, with an average rate of 5.2 GT yr$^{-1}$. In comparison, the annual mass loss from modelled freshwater runoff during the same period was 0.9 GT yr$^{-1}$, only 18% of the frontal ablation. The recent rates of frontal ablation are as high as comparable observations from the early phase of the surge a decade ago (4.2 GT yr$^{-1}$ for 2012-2013[21]) despite a considerable glacier deceleration since then.

The mechanisms by which ocean temperature affects glacier termini can be complex and may involve subglacial runoff or meltwater plumes entraining warm ambient water[13,32], feedback processes between glacier front processes and local ocean circulation[10,33,34] and impacts on calving dynamics through e.g. undercutting[35–37]. Our approach does not distinguish between frontal ablation in the form of melting and calving of solid ice, and it remains unclear exactly how thermal ocean forcing affects the glacier termini. However, we find that ocean heat is relatively more important than subglacial runoff for frontal ablation of Storisstraumen. The greatest ablation rates were found during periods with warm ocean temperatures and some amount of freshwater runoff, but apparent ocean-driven ablation frequently occurred or even peaked during periods of no modelled runoff. This could mean that ocean heat is a necessary condition for frontal ablation, while subglacial runoff may act to enhance or to initiate the seasonal frontal ablation. This is broadly in line with two studies from western Svalbard fjord sites[38,39] that both showed the highest retreat of tidewater glaciers during periods of *both* elevated surface melting and warm ocean temperatures.

## Warm inflows and surface heating drive seasonal ocean warming
Ocean temperatures at the ocean mooring underwent a strong seasonal cycle, broadly alternating between near-freezing temperatures in spring to summer and warmer temperatures (>0 °C) in autumn and early winter, but with considerable interannual variability (Fig. 2c). Two

main mechanisms may cause seasonal ocean warming in the depth range of the Storisstraumen calving front (0–100 m, Fig. 1e); heat transfer across the air-sea interface and inflow of relatively warm modified Atlantic Water.

Intensified warming at depth during late 2018 and 2019 at the mooring site has been attributed to the seasonal inflow of Atlantic-influenced water from the boundary current region southward into the Barents Sea[17]. In 2020 and 2022, ocean warming observed at the mooring was intensified near the surface, consistent with high air temperatures and low sea ice extent in the region during these years (Fig. S1) – both promoting increased warming near the ocean surface. Although shallow and deep warming both tended to peak in autumn, they occasionally occurred out of phase, like in 2021. This reflects interannual variability in environmental factors like atmospheric temperature and net downward radiation, sea ice concentration, and ocean advection of heat from the north.

The ocean mooring location is just upstream of the expected flow pathway of the coastal circulation (Fig. 1a). An open pathway at ~100 m depth between the ocean mooring and the Storisstraumen terminus allows a branch of the southbound current to reach the glacier face. Measurements collected during a shipboard hydrographic transect from February 2021[40] indicate the presence of relatively warm (>0 °C) waters at depth in a deep (>100 m) trough extending from the mooring location toward Storisstraumen and Hartogbukta bay at the northern end of the glacier front (Fig. 3a, c). Current measurements from the same transect (Fig. 3d) are consistent with the circulation of deep warm water masses toward land in these troughs. Temperature profiles taken in inner Hartogbukta (Fig. 3b) show the presence of deep, warm waters at depth close to the coast during cold and sea ice-covered ocean conditions.

These ocean observations provide evidence for transport of deep warmer waters from the coastal ocean toward the glacier front, although we do not know how frequently or in what magnitude this transport occurs. Wind, sea ice melt and formation, and tidal forcing could conceivably all play a role in mediating the nearshore ocean circulation. We take the temperatures observed at the mooring as representative of the seasonality of the coastal ocean more broadly—not as the exact water temperatures at the glacier face. The greater correlation with frontal ablation at 20 m than at 95 m depth (Fig. S3) was due in part to periods where the deep ocean at the mooring site was relatively warm while the glacier frontal ablation was low or near zero, like the winter of 2021-2022 (Fig. 2). This could be a result of the weak circulation of warm deep waters to the front during these periods, or due to a more local deep and cold winter mixed layer comprising the bulk of the ambient water column at the glacier face.

## Seasonal terminus retreat along the eastern and northern Austfonna margin
A seasonal advance-retreat cycle consistent with largely unobstructed ice flow in winter and ocean-forced retreat due to frontal ablation in autumn is evident both along the ~150 km eastern calving front of Austfonna (including Storisstraumen), and at outlet glaciers in the north which drain into small, enclosed embayments (Fig. 4). The frontal ablation of Storisstraumen alone was comparable to that of the other 18 marine-terminating glaciers on Austfonna put together, with a striking similarity in timing and pattern of frontal ablation between glaciers in the north, east and south sectors during 2016-2021 (Fig. 4c). Glaciers in the western sector have lower frontal ablation rates and show little systematic seasonality, possibly due to their more shielded oceanographic setting at the end of two long fjords (Fig. 4). Also noteworthy is the near lack of frontal ablation in summer-autumn 2019 when the near-surface water at the ocean mooring remained cold. This was despite a build-up of warm water at depth which peaked in December coincident with a brief spell of frontal ablation at Storisstraumen (Fig. 2).

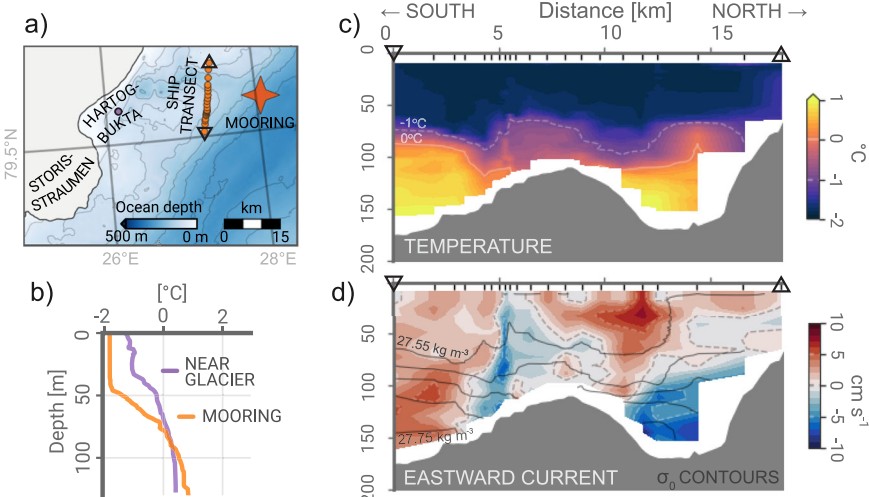

**Fig. 3 | Shipboard survey of the coastal ocean.** Shipboard ocean measurements from February 2021. **a** Map of measurement stations. **b** Temperature profiles near the M1 mooring and in Hartogbukta near Storisstraumen glacier. **c** Temperature and **d** detided zonal current component from cross-transect between M1 and Storisstraumen. Black solid contours in d show lines of constant $\sigma_0$ density.

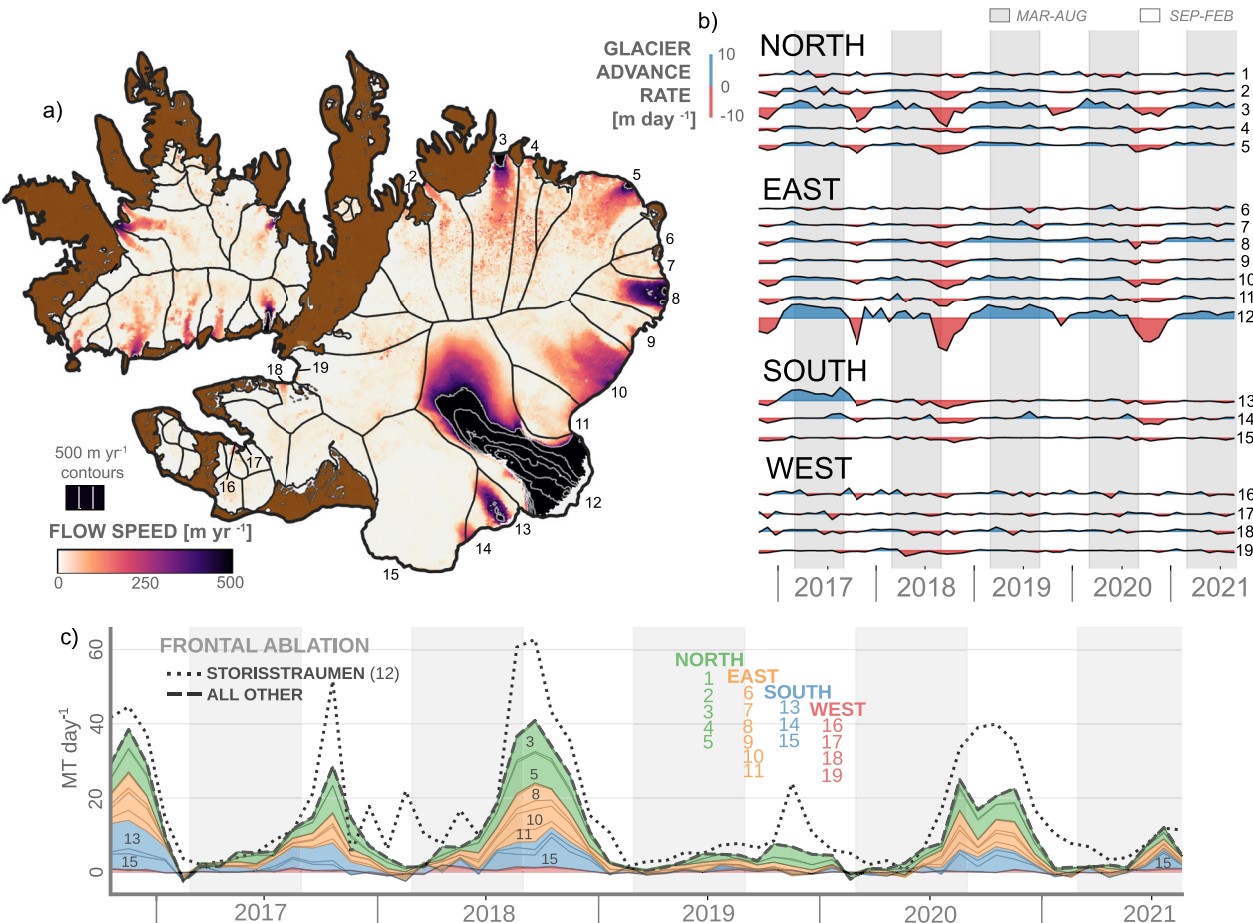

**Fig. 4 | Glacier flow speed and glacier front time series. a** Mean glacier flow speed during 2018 from the ITS_LIVE dataset[59]. **b** Advance rates of marine-terminating glacier fronts of Austfonna, Oct 2016 to Sep 2021. **c** Frontal ablation of the same glaciers. Notable named glaciers: Schweigaaardbreen (3), Leighbreen (5), Storisstraumen (12), Klerckbukta (13), Bråsvellbreen (15), Etonbreen (19). Glacier basins extracted from the glacier inventory of Svalbard[70]; note that the Klerckbukta-Kervelbukta basin (13 and 14) was split into two units due to their separated ice-flow and surging activity.

Common to the glacier fronts that exhibit a strong seasonal advance-retreat cycle due to ocean forcing are i) a deep grounding line depth and ii) a relatively high ice flow velocity. The first means that warm water at depth can access the glacier base. The second may imply that more dynamically active glacier fronts are more responsive to ocean forcing. A fast-flowing front is likely to have higher strain rates, forming fractures and crevasses that can propagate into calving ice blocks, especially in combination with undercutting from submarine melting (e.g[41]).

Unlike the western and southern coast of Nordaustlandet, much of the nearshore bathymetry of the northeastern side of the island is poorly mapped (Figs. 1e, S2), and we do not know whether deep throughs connect the offshore deep waters with glacier fronts in this area. However, troughs connecting deep offshore basins with nearshore areas are present along large parts of the southeastern and northern margins where bathymetry data are available, indicating that there are potential pathways of warm offshore waters toward much of the Austfonna calving front.

## Ocean-driven glacier mass loss in changing high Arctic regions

The northern Barents Sea has experienced exceptional rates of atmospheric warming[2] and rapid sea ice loss[3] during recent decades, making the region a global hotspot of environmental change[42]. From the mid-2000s, the northern Barents Sea experienced ocean warming both near the surface and at mid-depth[4], a regional expression of what has been termed the Atlantification of the Eurasian sector of the Arctic Ocean[5,43]. This period coincides approximately with observations of increased mass loss from glaciers on the Barents Sea margin of Svalbard[44]. The rapid response of Storisstraumen and other active calving fronts to seasonal fluctuations in ocean temperatures confirms that the northern and eastern marine-terminating glacier margin of Svalbard is highly sensitive to ocean warming. We hypothesize that the glaciers will retreat and lose mass even faster if Atlantification and warming of the northern Barents Sea continues. In particular, ocean-driven melting that occurs near the grounding line may affect basal friction and the force balance near the terminus, which in turn may result in acceleration of the glacier and cause dynamic thinning of upstream areas. The deep bedrock and large fringe areas of the eastern part of Austfonna (Fig. 1c) means that glacier termini can potentially continue to retreat for tens of kilometres before they eventually lose connection with the ocean.

Canonical theory for discharge-driven melt at tidewater glacier termini predicts a glacier front melt rate that is proportional to the ocean thermal driving, and to the cube root of subglacial runoff[45,46]. While the data and methods presented here focus on broad seasonal covariances than quantitative relationships, they do provide qualitative insights relevant to glacier-ocean interactions more broadly. Our findings indicate that i) ocean forcing is the primary driver of frontal ablation at Austfonna, and ii) subglacial runoff has limited impact on frontal ablation when the thermal driving is very small. This study is thereby in line with studies showing the impact of the ocean on mass loss from the world's glaciers (see, e.g[37,47,48]), including the response to seasonal ocean warming observed in western Spitsbergen[23,38,39].

In apparent contrast to these results, case studies from fjord sites in southwestern Greenland[49] and Alaska[10] have indicated that runoff discharge is the primary driver of seasonal glacier retreat, with variations in ocean temperatures playing a minor and/or secondary role. A potential reason for this apparent discrepancy is the large seasonal range in thermal driving in the northwestern Barents Sea compared to many well-studied systems in Alaska and Greenland where ocean temperatures below the mixed layer do not typically reach near-freezing seasonal minima (e.g.[49–51]). As a result, the relative seasonal range of thermal driving in these systems may be much smaller than in the northwestern Barents Sea, and therefore have less of an impact on seasonal variability in frontal ablation (see[10]). Secondly, most comparable studies focus on fjord systems where the constrained geometry leads to a largely two-dimensional circulation, with entrainment of ambient water into a buoyant plume driving warm ocean waters towards the glacier face[11,52]. This mechanism may be less important at the open geometry along most of the Austfonna marine margin, where one might suspect runoff-driven circulation to be more three-dimensional. Lastly, the relatively simple theoretical framework of plume-driven melt does not capture potential multiplying effects due to undercutting, which may play out differently in different physical environments.

The results of this study imply that eastern and northern Svalbard, the margins of the archipelago facing the Barents Sea and the Arctic Ocean, are regions where glaciers are highly sensitive to ongoing changes in the thermal forcing by the surrounding ocean. Atlantification and sea ice loss are predicted to progress further northward and eastward into the Arctic Ocean[53,54]. Glaciers and ice caps in other high Arctic areas, such as Novaya Zemlya, Franz Josef Land, and Servernaya Zemlya, are likely to be similarly susceptible to profound changes in the physical environment due to decreased sea ice cover and increased Atlantic Water presence in the future. Given that ocean forcing is not included in current region-scale glacier models[55], this development may cause higher than projected mass loss and sea level contribution from high Arctic glaciers.

## Methods
### Glacier observations

Advance (or retreat) rates along the marine-terminating margin of Austfonna were calculated from spatial intersection of glacier front lines manually digitized from synthetic aperture radar (SAR) imagery from Sentinel-1 at a resolution of 5 meters and at intervals of 24 to 36 days over the period 2016-2021[56]. All the images were pre-processed within the Sentinel Application Platform (SNAP) using a standard workflow[57]. Terrain correction was done with the NP S0 Digital Elevation Model (DEM) from the Norwegian Polar Institute[58]. Frontal area changes between successive images were computed for 19 drainage basins (Fig. 1a) and assigned a time stamp midway between the image dates. Glacier-averaged advance rates (m day$^{-1}$), negative for retreat, were obtained by dividing the area change by the length of a "fluxgate line" that was digitized just inland of the most retreated front position during the period. This approach ensures consistent advance rates from period to period rather than a dependency on the complexity/length of each front line. The front line time series for Storisstraumen was extended to autumn 2022 in order to cover the entire period of available ocean observations.

Glacier flow velocities were estimated for the fluxgate (resampled to a vertex point spacing of maximum 50 m) at the same time steps using the velocity products of the ITSLIVE[59] dataset, which consists of 2D velocity maps from almost any image pair combination of Landsat-8 (optical), Sentinel-1 (SAR) and Sentinel-2 (optical), in total thousands of velocity maps for Austfonna with variable coverage and quality. As a balance between robustness and temporal resolution, we selected velocity data from image pairs with >5 days and <30 days separation. For each time step, we used the median velocity of these data from within ±14 days, roughly corresponding to the time resolution of the front-mapping.

To obtain mass rates from velocities and front position changes, we need to consider ice thickness and density along the fluxgate for the areas of advance/retreat. We made a grid of ice thickness by subtracting a bed elevation model from a surface elevation model. Bedrock topography was obtained from the SVIFT 1.1 dataset[60] (Fig. 1c), extended with historical echo-sounder bathymetry data[17] for the surge-advance area of Storisstraumen. Surface topography was extracted annually from ArcticDEM strip-data[61], using a median approach to all data strips from each year of concern, after bias-removal against land areas of the NP S0 DEM, which is based on traditional aerial photogrammetry.

The ice thickness grid was combined with front area change and an ice density of 0.917 kg m$^{-3}$ to obtain frontal advance rates in terms of mass ($A$, MT day$^{-1}$) for each time step and glacier basin. Ice discharge to the front area ($D$, MT day$^{-1}$) was obtained by multiplying ice thickness with the density of ice, the ice-flow velocity vector (across the fluxgate) and the length of each line segment along the fluxgate. These fine-scale mass fluxes (<50 m spacing) were then summed up for each glacier basin and time step for total discharge estimates. Total frontal ablation $FA$, or the mass loss at the front due to melt and/or calving, was finally estimated as:

$$FA = D - A \qquad (1)$$

### Ocean observations

An ocean mooring, M1, was located approximately 50 km from the Storisstraumen glacier front (Fig. 1a). We show data from four back-to-back deployments of the mooring, between 5 October 2018 and 7 October 2022, with a data gap between 21 September 2020 and 20 February 2021. A previous study[17] using data from the first two deployments showed that the mooring was located in a seasonally varying slope current where warm water masses originating in the Atlantic Water boundary current north of Svalbard are transported south through Kvitøya Trough in late autumn to early winter (Fig. 1b). In this study, we use temperature sensors at ~20 m (deployment median depths 21, 20, 21, 22 m), ~60 m (60, 59, -, 59 m) and ~95 m (95, 97, 97, 92 m) depth as an indication of the evolution of the offshore ocean conditions through the study period. The instruments at ~60 m depth before autumn 2021 were temperature-only instruments without onboard pressure sensors, and the depths are therefore estimates. Depths of other sensors were calculated from observed pressure.

Profiles of hydrography (Conductivity-Temperature-Depth; *CTD*) and currents (Lowered Acoustic Doppler Current Profiler; *LADCP*) were collected from the R/V Kronprins Haakon in the area between M1 and Storisstraumen on 20-23 February 2021. CTD profiles were collected both near M1 and in Hartogbukta bay near the northern part of Storisstraumen glacier front. In addition, a transect across potential shoreward flow pathways was conducted by letting the ship drift along with the ice over a 27-h period, resulting in an 18 km northward high-resolution transect. LADCP currents were detided using the Arc2km tidal model[62].

### Freshwater runoff

Modelled freshwater runoff was obtained from the *CryoGrid* community model, version 1.0[63]. The model was forced using CS3 Arctic regional reanalysis, *CARRA*[64] during 2018-2020, and *AROME-ARCTIC* reanalysis[65] during 2020-2022. The runoff simulations include the delay of runoff due to percolation through snow and firn, and a runoff timescale based on the surface slope was used. However, the distance from the runoff point to the ocean was not included in the simulations. Details of the model simulations can be found in[27]. Runoff outlets (Fig. 1d) were calculated using *TopoToolbox*[66]. The outlets were calculated assuming subglacial flow by using the hydraulic head as model input with a flotation fraction of 0.9. High-resolution DEMs of the surface[58] and bedrock[60] elevation were used to calculate the hydraulic head. A time series of runoff from Storisstraumen was obtained by summing the volume flux over a total of 6 outlets exiting along the glacier front.

### Correlation analysis

We performed a simple, first-order quantitative analysis of the relationship between environmental variables and frontal ablation at Storisstraumen after first resampling all variables to daily resolution (linearly interpolating between adjacent points of coarser spacing) and then computing average values for each month from Nov 2018 through Sep 2022. From the monthly values, we computed zero-lag Pearson correlation coefficients and associated p-values were and linear fit parameters (Fig. S3). Months with missing data, such as during the late 2020 data gap in ocean temperature, were excluded from the correlation analysis.

### Data availability

The mooring data used in this study are available in the Norwegian Polar Data Centre database under accession code https://doi.org/10.21334/npolar.2022.1a68b156 (2018-2021) and https://doi.org/10.21334/npolar.2024.bb2d725f (2021-2022). The shipboard data used in this study are available in the Norwegian Marine Data Centre database under accession code https://doi.org/10.21335/NMDC-1544015310. The monthly glacier front line data used in this study are available in the Norwegian Polar Data Centre database under accession code https://doi.org/10.21334/npolar.2024.8ad491fc. The modelled freshwater runoff data used in this study are available in the Norwegian Polar Data Centre database under accession code https://doi.org/10.21334/npolar.2024.87fa4a0a. Other, publicly available datasets used in figures and analysis are cited throughout the manuscript and listed in the reference list below.

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

## Acknowledgements

The collection of ocean data and the work done by Ø.F., L.S.S., I.F., F.N., and A.S. were funded by the Nansen Legacy project (RCN 276730). J.M. was funded by the project Copernicus Glacier Service through the Norwegian Space Agency. D.S. received partial support from US NSF grant 2020447 from the Office of International Science and Engineering.

## Author contributions

Ø.F. and G.M. wrote the initial draft of the manuscript. Ø.F. produced the figures. J.M. extracted the glacier front lines from radar images. G.M. computed time series of frontal ablation and other quantities. L.S.S. conducted the runoff modelling. I.F. and F.N. collected and analysed the cruise transect data, and A.S. and Ø.F. the mooring data. Ø.F., J.M., G.M. and D.S. contributed to developing the original idea. Ø.F., G.M., J.M., L.S.S., I.F., F.N., J.K. and A.S. contributed to the writing of the manuscript.

## Funding

## Competing interests

The authors declare no competing interests,
