## [Transparent Peer Review file · Nature Communications]

Ocean warming drives immediate mass loss from calving glaciers in the high Arctic

Corresponding Author: Dr Øyvind Foss

Version 0:

Reviewer comments:

Reviewer #1

(Remarks to the Author)

This is a relatively simple but effective study illustrating the sensitivity of glaciers of the Austfonna ice cap to the temperature of the surrounding ocean. Controls on the rate of frontal ablation (i.e. calving plus submarine melting) at tidewater glaciers are an area of much interest in glaciology at present, given the importance of frontal ablation as a source of mass loss and sea level rise (particularly regarding the Greenland Ice Sheet). Key challenges faced in this area have been determining the importance of submarine melting as a driver of frontal ablation and terminus retreat, and constraining the key controls on submarine melting itself. By focussing on outlets from the Austfonna ice cap in Svalbard, this study is able to neatly demonstrate that in this location at least, submarine melting really does matter, with rates determined primarily by ocean temperatures rather than runoff input. This allows the authors to hypothesize that glaciers across the wider region will prove highly sensitive to the ongoing warming of the ocean in this sector.

The study site in Svalbard allows these conclusions to be reached because for several reasons. Firstly, frontal ablation rates are generally lower than at their larger, deeper, faster flowing Greenlandic counterparts, and with a clear seasonal cyclicality. Secondly, the ocean temperature shows a strong seasonal cycle of ~ 4 °C from close to the in situ freezing point to $\sim +2$ °C. Thirdly, and perhaps most novelly, the seasonal peak in ocean temperature occurs at a distinct time to the seasonal runoff peak, allow these competing influences to be separated. Fourthly, similar seasonal patterns are exhibited across glaciers throughout the area, illustrating the regional nature of the forcing. For these reasons this paper is able to demonstrate with an unusual degree of confidence that the frontal ablation rate (of these glaciers at least) is controlled primarily through ocean temperature via its influence on submarine melt rates.

It's worth noting that while these results are convincing and certainly make a valuable contribution to the literature, there is similarity with a paper published in Nature Communications in 2015 by Adrian Luckman and others, which used data from three other glaciers in Svalbard to similarly demonstrate that ocean temperatures were a strong control on frontal ablation rates. The current manuscript goes beyond this in valuable ways, particularly (a) more clearly separating out the impacts of runoff and (b) demonstrating applicability to a wider range of glaciers, but the development on the early paper is perhaps incremental rather than representing a major step up in our understanding.

One area in which the paper could be strengthened a little is through greater discussion of the broader applicability of the findings and their contribution to our understanding of these processes. For example, a recent paper on a glacier in Alaska (Jackson et al, 2022) concluded that there was no discernible impact of ocean temperature on submarine melt rate in that location, with runoff the dominant control (the inverse finding of this manuscript). This paper is cited, but there is no comment on this disparity or what it may tell us about submarine melt mechanisms. Can the authors go further in using their excellent data set to test or improve theories on submarine melt rate? How do submarine melt rates (modelled using theory such as proposed by Jackson et al 2022) match with observed frontal ablation rates (which serve as a maximum upper limit for submarine melt rates) over the temperature range observed in this study? Greater analysis and discussion on topics such as these would help this paper to go from one with a predominantly regional focus to one able to make a greater contribution to the general theory on the subject and thus with more widely impactful findings.

In summary, this is an excellent observational study that neatly and convincingly demonstrates the importance of ocean temperatures for tidewater glaciers in Svalbard's glacial, climatic and oceanic regime. It strengthens the findings of the earlier paper by Luckman et al (2015) by allowing a clear separation of the impacts of runoff and ocean temperatures, and

the findings will be of interest and value to the glaciology community. There is however potential to widen the significance of the findings through greater discussion of the broader applicability of results, and through the use of the data to contribute to theory on the subject that can be used to understand and predict the behaviour of tidewater glaciers more broadly.

Beyond these general reflections, I have only a few specific comments. The manuscript is well written and the conclusions are robustly supported by the evidence presented.

L52-53 and elsewhere. There is not presently much background on submarine melting and its controls, and so it's not clear from the paper why freshwater runoff might be expected to influence submarine melt rate.

Figure 2. Could we also see frontal ablation rates in metres per day for ease of comparison with other papers and the common unit of submarine melt rates?

L69-70. I'm not sure this can be ascribed to the ocean from this evidence alone. It's likely that the glacier was larger and thicker during the 19th Century, resulting in a more extensive surge irrespective of the ocean conditions.

L107-108. Can this relationship be investigated more thoroughly, by looking statistically at the combined and separate influence of ocean temperatures and runoff on frontal ablation rate?

L109-110. It's not clear to me what is meant by an 'initiator' here. Do you mean that runoff may enhance melt rates, but that there can be little melt (irrespective of runoff) if the temperature is close to the in-situ freezing point?

L148. The term 'winter of 2022' is ambiguous – consider 'winter of 2021-22' or give specific months.

L156. Although the amplitude of variability at these glaciers is clearly small, it does look like there are some similarities between glaciers and with other regions, such as during the 2018 retreat. How do they compare with the other regions if trends for each glacier are normalised based on the range exhibited at that glacier?

L173. Bråsvellbreen is defined as in the south, rather than southeast, in Figure 4.

Reviewer #2

(Remarks to the Author)

The authors present a time series of (ocean temperature, glacier speed and retreat) observations of marine-terminating glaciers in eastern Svalbard that suggests that the glacier retreat is in great part controlled by the ocean. I find the paper of interest, well written, useful, but it would be great if the authors could clarify what is observation-based and what is hypothesis-based, i.e. they don't actually have the measurements to say so. The paper is also not well connected with research in Greenland, very Svalbard focused, but the processes discussed are universal, the main difference is that ocean forcing is quite big in Svalbard, so these data are important sentinel of change.

I do recommend publication after minor revision.

Line 31: Where are the marine terminating glaciers? this says that there are important in Svalbard, but Fig 4 shows that they are mostly in eastern Svalbard. On the west, presumably because of warmer waters (any documentation of that would be useful to add), the glaciers are already detached from the ocean, and perhaps glacier depth is higher on one side??

Figure 1a. How much of the bathymetry around Svalbard is real data versus interpolation? Do you have enough measurements in front of the glaciers to make sure you do not miss troughs? If there are regions with data gaps, it would be useful to indicate that the seafloor in this gappy areas is likely to be deeper than represented in IBCAOv3.

Figure 1c. Please make sure to delineate regions where bed > sea level and regions where bed < sea level. The current color coding does not make that easy. There should be a drastic change in color from above to below msl to make it easy for the reader to find the marine-based sector.

Line 48. Do not state your conclusions in the intro. Perhaps say "we evaluate the impact of ocean forcing".

Line 69. I do not think that this statement is justified.

Line 74. Can you show ocean data that document that, for instance with a mooring on the west or CTD data? How much warmer on the west?

Line 97. Is it understood that these glaciers calved at the grounding line and do not have any floating section? If so, how do you know and can you please clarify?

Line 104. It would be easy from these studies of undercutting to estimate the rate of undercutting from depth and thermal forcing for your glaciers. Then you could compare with ice velocity and with a rate of undercutting calculated in more "normal" ocean conditions (perhaps much older ocean temp data?). This would help build the quantitative nature of the

manuscript which is currently more based on temporal correlation than actual numbers.

Line 105. Unclear but you have not even tried.

Line 116. Typo, there is no Fig. 2d.

Line 120. This could come earlier on. In Greenland, the warm water is found deeper. 100 m is shallow, but you have data that show it. The water that matters is at the grounding line depth of the glaciers. I am not sure that we need to know the temperature at 20 m depth other than to provide contextual information.

Line 139. Again, do you have proper knowledge of ALL the troughs?

Line 158. Deep grounding line means more thermal forcing and more access of warm waters (> 100 m) to the glacier base. The authors seem to think that areal melt matters, but that's not true. The melting that matters is the melting near the base, because it will affect basal friction. Melting in intermediate layers do nothing much to the glacier force balance, unless you can explain that to the reader.

Line 161. You may be right but I don't what this is based on. Speculative? A fast front calves more because .. it has to.

Figure 4. So these are the big guns that are marine based. They only show up in Figure 4. Could you mention that earlier?

Line 171. Did you document surface runoff vs basal melting of grounded ice from geothermal and friction? It would be good to see these numbers if you have them. Not too hard to get, but no numbers in the paper.

I found the whole paragraph rather speculative.

Line 178. Sea ice is important as it modulates the production of salty, cold water which might modulate the incursions of Atlantic water to the glaciers. This "making" effect is not mentioned, but it might be of interest to quantify the sea ice cover in front (to be defined) of these glaciers. Numerous papers have shown this to be an important factor. Could you at least mention that?

Line 189. But is not it true that this is the only part of Svalbard with marine terminating glaciers? In that case, not a big surprise if the west does not follow the ocean ..

Line 196. Please add references that show that ocean forcing is not included in the projections of Svalbard glacier evolution. I suspect it is true, but quoting a reference would make this statement stronger.

Overall this is a great study. I would encourage the authors to make one more step - to the extent possible - to be more quantitative with melt numbers, warm water numbers (e.g. any evidence for long term warming on the east side?), estimates based on simulations would be better than nothing. So many papers make the claim that the ocean matters, they have the ocean forcing, the end result (retreat), but do not try to connect the numbers. Why? If it is hard to do, say so, and mention that loud and clear, and then make sure to collect more data (e.g. on melt rates of grounded ice) in the future; otherwise accept - and state - that the inference is only a temporal correlation at this stage. That's important to state.

Reviewer #3

(Remarks to the Author)

Dear Manuscript Authors and Nature Communications Editors,

In this study, Foss et al investigate drivers of ice front variability at glaciers draining the Austfonna ice cap, finding that oceanic (rather than atmospheric) temperature variability is responsible for modulating glacier ablation. The authors leverage a unique mooring dataset spanning a few years in addition to remotely-sensed data of advance/retreat and ice velocity. While many of the glaciers on the ice cap are considered, the main focus and majority of the study discusses Storstraumen glacier.

In reading this manuscript, I found the study to be a robust interpretation of the data and an interesting addition to a broader narrative of studies linking ice loss and sea level rise to the "Atlantification" of the Arctic. The "story" is clear, as are the figures and analysis of data.

However, the scope of the study is quite limited – the study is focused on a handful of glaciers (mostly one) on a single ice cap in the Arctic for a timespan of a few years. The temporal and spatial scope is largely based on the time duration and location of the available mooring data. I acknowledge that maintaining (and funding!) a mooring in this location is quite a difficult pursuit but it should be noted that the mooring data is not novel to this study as the lead author has already published a separate manuscript on this data (Lundesgaard et al 2022).

In considering this paper for Nature Communications, I am not sure the scope is sufficient as it's written. Other papers on the same topic published in this journal tend to be extended in time (e.g. Millan et al 2023) and/or extended in space to be pan-Arctic (e.g. Kochitzky et al 2023). Given this precedent, I would recommend this study for publication in this journal if the observational results were leveraged to analyze a longer time period (decades) and/or potentially more glaciers (at least all

of Svalbard/Austfonna). There are decades of available remotely sensing data which could be used to quantify glacier evolution. Similarly, a comparison of the mooring data with available ocean state estimates (e.g. from NEMO, ECCO, etc) would likely allow an extension of the analysis over similar periods and large spatial areas using model output. These two records would provide a valuable extension and would likely raise the impact of this study significantly.

Overall, I would recommend this paper for publication after the major revisions described above. If that path is taken, I would be more than happy to review a revised version of this study.

Sincerely,

Mike Wood

Version 1:

Reviewer comments:

Reviewer #1

(Remarks to the Author)

Thanks to the authors for addressing the points raised in my earlier review. I think that these have strengthened the manuscript, particularly through some considered discussion of the new findings in the context of existing studies. I had only a few specific comments on the earlier manuscript and these have to a large extent been addressed effectively. I have just a few minor points of clarity on the current version.

Comments:

L78-79. As I think I flagged in my previous review, the early part of the manuscript lacks explanation of the mechanisms that might be expected to affect frontal ablation. As such, if you're not already versed in the subject, it's not obvious at this point why one might expect meltwater runoff to be a major control on frontal ablation. Meltwater plumes are mentioned later (e.g. L140), but I think it would be helpful to have a very brief explanation earlier in the paper of why meltwater discharge has been hypothesized as a key control, given it is a key focus on the study.

L200. Thank you for providing the additional figure on this point. I agree that the western glaciers show little obvious seasonality, but I do think they show some synchronicity with each other and the other sectors with respect to periods of major retreat or advance (e.g. the major retreat phase during mid 2018). I'm not going to demand that you include this point, but I do think the data from these fjords supports your broader hypotheses more than you allow for in your description.

Figure S3. It's striking from this figure and the computed correlations that the correlation between frontal ablation and ocean temperature is much stronger at 20 m than at 95 m (which is no better than air temperature as a predictor). I think this merits a short discussion, not least to counter any criticism that the conclusions are overly sensitive to which depth you happen to select for water temperature. Do you think there is a physical explanation for this, or is it due to data quality, or something else?

Figure S1a. Having noted the correlation between air temperature and frontal ablation in Figure S3, it would be useful if this was plotted alongside the other main timeseries in Figure 2 (or if these were added to Figure S1). It's not possible to meaningfully compare the timing of seasonal trends between the two figures as it stands.

Reviewer #2

(Remarks to the Author)

I am satisfied with the revision, except for one point that I wanted to clarify again. Apart from this point, which the authors are welcome to take into account or not, I am ok with the publication of the paper. I do not need to see their response.

The authors state "We do not entirely follow the reviewer in that areal melt does not matter. In glaciers where terminus melt is important, the submarine melt itself is certainly large enough to be a major factor in the glacier mass balance. Melting in intermediate layers may still cause undercutting (see our response to 2.11 above). And while the glacier force balance near the terminus may play into the process in several ways, it is not necessary to consider glacier dynamics in order to explain frontal ablation driven by ocean thermal forcing."

In terms of ice melt, yes, frontal melting matters, but as long as the ice crosses the (fictitious) grounding line, it does not matter if ice melts or breaks up in the ocean, it will contribute to sea level rise. More important is the effect of melt on the glacier stability, because this will determine if the glacier will speed up or not in response to an increase in melt. Frontal melt distributed along the ice face does not matter in the force balance, while melting at the grounding line matters greatly. If you remove grounded ice, you reduce basal friction, which effects the glacier force balance. Conversely, melt at the water line

does not matter, it leaves the majority of the ice column below water friction at the bed. The ice above it, that's typically about 10% of the ice thickness. If you melt ice right at the grounding line, however, the entire column of ice above the cut will no longer affect basal friction, which is the intrinsic power of undercutting

I do not know if I explained this well enough, but I hope the authors will give it some thought. It is important to support the point that the shallow water temperature do not matter; the one that matters is the water temperature at the depth of the grounding line.

END

Reviewer #3

(Remarks to the Author)

Dear Manuscript Authors and Nature Communications Editors

The revised manuscript of "Ocean warming drives immediate mass loss from calving glaciers in the high Arctic" by Føss et al has undergone significant improvements since the initial draft. Here, I will briefly summarize my initial comments and my impressions on the responses from the authors.

My primary comment on the initial draft was that the paper was limited in scope relative to other papers submitted to Nature Communications. This viewpoint was echoed in some of the comments provided by the other reviewers on the manuscript. To expand the scope, I suggested that the study could be extended to all glaciers in Svalbard, and that the authors could leverage historical imagery and ocean reanalysis products to extend the study backwards in time.

For the number of glaciers in Svalbard, the authors have expanded their study to all of the glaciers, now providing additional timeseries in the revision. This has provided a significant improvement to the scope of the study.

For the extension backwards in time, the authors conducted a comparison between the mooring and reanalysis products in the region and have even provided some figures for the comparison. The authors have concluded that the reanalysis products are not trustworthy at depth (100m) and therefore that the extension backward in time is not possible. In looking at the timeseries however, the reanalysis products do seem to be able to capture interannual variability at depth even though they do not get the absolute magnitude quite right. I would imagine that if the timeseries were extended backward in time, the "Atlantification" mentioned in this paper would be visible in both temperature and salinity. Further, since the paper has already linked ocean temperature with the glacier retreat, I would suspect the glaciers would retreat in concert with the warming trend, and that finding would significantly bolster the impact of the study. However, I acknowledge that this would entail a significant amount of additional work, primarily in the extension of the frontal ablation timeseries.

Given these considerations, I will defer to the editor and other reviewers for the decision for publication. I think the tools are in place to generalize from mooring to ocean reanalysis and this would provide valuable information over several decades during a time of widespread change in the Arctic. However, I also acknowledge that this would require additional work and may be addressed in an additional separate study.

Sincerely,
Mike Wood

RESPONSE TO REVIEWERS, NATURE COMMUNICATIONS

This document contains the response to the reviews received for our submitted manuscript *Ocean warming drives immediate mass loss from calving glaciers in the high Arctic*. The original comments from the three reviewers are shown in black, with our numbering for easy reference. Author responses are shown in blue.

Line numbers refer to the track-change PDF document.

Reviewer 1:

This is a relatively simple but effective study illustrating the sensitivity of glaciers of the Austfonna ice cap to the temperature of the surrounding ocean. Controls on the rate of frontal ablation (i.e. calving plus submarine melting) at tidewater glaciers are an area of much interest in glaciology at present, given the importance of frontal ablation as a source of mass loss and sea level rise (particularly regarding the Greenland Ice Sheet). Key challenges faced in this area have been determining the importance of submarine melting as a driver of frontal ablation and terminus retreat, and constraining the key controls on submarine melting itself. By focussing on outlets from the Austfonna ice cap in Svalbard, this study is able to neatly demonstrate that in this location at least, submarine melting really does matter, with rates determined primarily by ocean temperatures rather than runoff input. This allows the authors to hypothesize that glaciers across the wider region will prove highly sensitive to the ongoing warming of the ocean in this sector.

The study site in Svalbard allows these conclusions to be reached because for several reasons. Firstly, frontal ablation rates are generally lower than at their larger, deeper, faster flowing Greenlandic counterparts, and with a clear seasonal cyclicality. Secondly, the ocean temperature shows a strong seasonal cycle of ~ 4 °C from close to the in situ freezing point to $\sim +2$ °C. Thirdly, and perhaps most novelly, the seasonal peak in ocean temperature occurs at a distinct time to the seasonal runoff peak, allow these competing influences to be separated. Fourthly, similar seasonal patterns are exhibited across glaciers throughout the area, illustrating the regional nature of the forcing. For these reasons this paper is able to demonstrate with an unusual degree of confidence that the frontal ablation rate (of these glaciers at least) is controlled primarily through ocean temperature via its influence on submarine melt rates.

It is worth noting that while these results are convincing and certainly make a valuable contribution to the literature, there is similarity with a paper published in Nature Communications in 2015 by Adrian Luckman and others, which used data from three other glaciers in Svalbard to similarly demonstrate that ocean temperatures were a strong control on frontal ablation rates. The current manuscript goes beyond this in valuable ways, particularly (a) more clearly separating out the impacts of runoff and (b) demonstrating applicability to a wider range of glaciers, but the development on the early paper is perhaps incremental rather than representing a major step up in our understanding.

One area in which the paper could be strengthened a little is through greater discussion of the broader applicability of the findings and their contribution to our understanding of these processes. For example, a recent paper on a glacier in Alaska (Jackson et al, 2022) concluded that there was no discernible impact of ocean temperature on submarine melt rate in that location, with runoff the dominant control (the inverse finding of this manuscript). This paper is cited, but there is no comment on this disparity or what it may tell us about submarine melt mechanisms. Can the authors go further in using their excellent data set to test or improve theories on submarine melt rate? How do submarine melt rates (modelled using theory such as proposed by Jackson et al 2022) match with observed frontal ablation rates (which serve as a maximum upper limit for submarine melt rates) over the temperature range observed in this study? Greater analysis and discussion on topics such as these would help this paper to go

from one with a predominantly regional focus to one able to make a greater contribution to the general theory on the subject and thus with more widely impactful findings.

In summary, this is an excellent observational study that neatly and convincingly demonstrates the importance of ocean temperatures for tidewater glaciers in Svalbard's glacial, climatic and oceanic regime. It strengthens the findings of the earlier paper by Luckman et al (2015) by allowing a clear separation of the impacts of runoff and ocean temperatures, and the findings will be of interest and value to the glaciology community. There is however potential to widen the significance of the findings through greater discussion of the broader applicability of results, and through the use of the data to contribute to theory on the subject that can be used to understand and predict the behaviour of tidewater glaciers more broadly.

Beyond these general reflections, I have only a few specific comments. The manuscript is well written and the conclusions are robustly supported by the evidence presented.

Thank you for the thorough and encouraging review. We particularly appreciate the suggestion to place our findings in a broader context (which is in line with similar points raised by Reviewer 2). In the revised manuscript, we have included two new paragraphs (L262-L284 in the track-change manuscript) dedicated to discussing our findings in the broader context of glacier-ocean interactions, as well as comparing and contrasting with previous studies.

We have made some additions in order to more quantitatively substantiate the connections (or lack thereof) described in the study (see R1.4 below). We have not attempted to evaluate the buoyant plume model explicitly since we lack the detailed, near-glacier measurements that would allow us to reasonably do so – there are simply too many unknowns to meaningfully contribute to constraining parameters in the model (we hope that this study can inspire more detailed studies and associated field campaigns in the future!).

The conclusions we can draw from our records are also somewhat more general, yet still relevant to the field more generally. In the revised manuscript, we generalize our findings and discuss possible reasons for apparent discrepancies between our study and others, including Jackson et al., 2022, pointed out by the reviewer.

As a result of these edits, we believe the revised manuscript has been significantly improved, and the contributions to wider questions in field strengthened.

R1.1

L52-53 and elsewhere. There is not presently much background on submarine melting and its controls, and so its not clear from the paper why freshwater runoff might be expected to influence submarine melt rate. several months after

Thank you for pointing out the unfortunate wording here. We indeed do not mean to imply that freshwater runoff influences submarine melt rate months later, but we see how the phrasing might make it seem that way. We have rephrased the sentence in question (L80-8)3 and the encompassing paragraph to make it clear that our interpretation of the time series is that freshwater runoff does *not* drive the submarine melt rate. New paragraphs near the end of the revised manuscript (L261-283) go into more detail about the role of subglacial discharge in this and other studies.

R1.2

Figure 2. Could we also see frontal ablation rates in metres per day for ease of comparison with other papers and the common unit of submarine melt rates?

Thank you for the suggestion. We have added a second y-axis to Fig. 2b showing approximate scale of frontal ablation in m/day (approximate because there is not a linear mapping between m/day and MT/day since the glacier height is not uniform).

R1.3

L69-70. I'm not sure this can be ascribed to the ocean from this evidence alone. It's likely that the glacier was larger and thicker during the 19th Century, resulting in a more extensive surge irrespective of the ocean conditions.

We agree that this could be read as if the glacier would otherwise have advanced to its historical surge extent, which is too speculative. We have rewritten this to instead focus on the present surge: *“Considering that the glacier has been in a surging fast-flow mode since 2012 (Dunse et al., 2015), we hypothesize that the glacier would have advanced much further into the Barents Sea if ocean conditions were colder.”* (L110-115).

R1.4

L107-108. Can this relationship be investigated more thoroughly, by looking statistically at the combined and separate influence of ocean temperatures and runoff on frontal ablation rate?

Thank you for the suggestion. We have underpinned the statements in question (and the visual illustration from Fig 2) with a quantitative analysis. We have attempted to keep the analysis relatively simple. Additions are found in:

- Scatter plots showing correlations and fits, new Fig S3.
- A brief description in the Methods section, L361-367.
- New content in the main text, L99-104.

R1.5

L109-110. It's not clear to me what is meant by an 'initiator' here. Do you mean that runoff may enhance melt rates, but that there can be little melt (irrespective of runoff) if the temperature is close to the in-situ freezing point?

We agree that this was phrased confusingly. The second part of this sentence, *“while subglacial runoff may act as an initiator”*, has been replaced with *“...may act to enhance or to initiate the seasonal frontal ablation”*. The start of the sentence (*“This could mean that..”*) makes it clear that this is speculation from our side. (L157-159).

R1.6

L148. The term 'winter of 2022' is ambiguous – consider 'winter of 2021-22' or give specific months.

Agreed, thanks for the suggestion (L202).

R1.7

L156. Although the amplitude of variability at these glaciers is clearly small, it does look like there are some similarities between glaciers and with other regions, such as during the 2018 retreat. How do they compare with the other regions if trends for each glacier are normalised based on the range exhibited at that glacier?

Thank you for the comment. We have looked into this; Appendix II below shows the retreat rates in Fig 4b with normalized y-ranges. The figures from the western sector are in our opinion in good agreement with the current phrasing, "*Glaciers in the western sector have lower frontal ablation rates and show little systematic seasonality [..]*".(L213-214)

R1.8

L173. Bråsvellbreen is defined as in the south, rather than southeast, in Figure 4.

Thanks for the suggestion. The sentence was removed from the manuscript in the course of other edits (L240).

Reviewer 2

The authors present a time series of (ocean temperature, glacier speed and retreat) observations of marine-terminating glaciers in eastern Svalbard that suggests that the glacier retreat is in great part controlled by the ocean. I find the paper of interest, well written, useful, but it would be great if the authors could clarify what is observation-based and what is hypothesis-based, i.e. they don't actually have the measurements to say so. The paper is also not well connected with research in Greenland, very Svalbard focused, but the processes discussed are universal, the main difference is that ocean forcing is quite big in Svalbard, so these data are an important sentinel of change.

I do recommend publication after minor revision.

Thank you very much for your comprehensive review of our manuscript. We appreciate your many constructive suggestions, which have significantly improved the quality of our study.

In response to your comments, we have clarified the distinction between observation-based and hypothesis-based statements throughout the manuscript. We have removed or revised some of the more speculative statements, and made it clearer how we draw our conclusions from our data and analysis. Additionally, following your suggestion and that of Reviewer 1, we have worked to broaden the geographical and theoretical focus of our results by incorporating a discussion of our findings in relation to theory and studies from other regions, such as Greenland and Alaska.

In response to several comments requesting more information about the physical environment, we have included a new section labelled "A major ice cap in a unique oceanographic setting" (L58-77 in the track-change manuscript) where we present the setting in a bit more detail before presenting the results of the study.

R2.1

Line 31: Where are the marine terminating glaciers? this says that they are important in Svalbard, but Fig 4 shows that they are mostly in eastern Svalbard. On the west, presumably because of warmer waters (any documentation of that would be useful to add), the glaciers are already detached from the ocean, and perhaps glacier depth is higher on one side??

Thank you for the useful comment. Marine-terminating glaciers in Svalbard are not in fact restricted to the eastern side of the archipelago, although this is where we find the long, uninterrupted marine margin. We agree that it is useful to set the stage for the reader a more thoroughly, and included a new section introducing the reader to the physical environment, including the glacier (L59-L64) and ocean (L65-L77) setting.

We have also chosen to show the Svalbard marine glacier fronts in the new Figure 1c. To the second point, the same panel also shows climatological ocean temperatures.

The new Figure 1c replaces the previous Figure 1d showing the subglacial drainage network. We realized that the latter was not adding anything particularly valuable for interpreting the study.

R2.2

Figure 1a. How much of the bathymetry around Svalbard is real data versus interpolation? Do you have enough measurements in front of the glaciers to make sure you do not miss troughs? If there are regions with data gaps, it would be useful to indicate that the seafloor in this gapy areas is likely to be deeper than represented in IBCAOv3.

Thank you for the suggestion to more clearly show which areas are missing good bathymetry data. In the revised Figure 1 (panel e), we have added hatched shading showing areas where there are notable gaps in the source data for the IBCAO bathymetry. A more detailed map of the IBCAO data type has also been included in the Supplementary information (Figure S2). We discuss the missing bathymetry data in a new paragraph (L242-247).

R2.3

Figure 1c. Please make sure to delineate regions where bed > sea level and regions where bed < sea level. The current color coding does not make that easy. There is should be a drastic change in color from above to below msl to make it easy for the reader to find the marine-based sector.

Thank you for the suggestion. We have updated the bedrock figure (new Fig 1d) in order to more clearly separate the regions above sea level from those below.

R2.4

Line 48. Do not state your conclusions in the intro. Perhaps say "we evaluate the impact of ocean forcing".

A sensible comment. We have changed the sentence along the lines of the reviewer's suggestion (L55-57).

R2.5

Line 69. I do not think that this statement is justified.

This was also pointed out by reviewer 1 (see R1.3. above), and we agree that this could be read as if the glacier would otherwise have advanced to its historical surge extent, which is too speculative. We have rewritten this to instead focus on the present surge: *"Considering that Storisstraumen has been in a surging fast-flow mode since 2012 (Dunse et al., 2015), we hypothesize that the glacier would have advanced much further into the Barents Sea if ocean conditions were colder."* (L110-115).

R2.6

Line 74. Can you show ocean data that document that, for instance with a mooring on the west or CTD data? How much warmer on the west?

Thank you for the suggestion. In order to show the difference between east and west, we have included a figure (Fig 1c) showing mean ocean temperature at 50 m depth from the NOAA Arctic Ocean Regional Climatology (Boyer et al., 2015). Although such a product has its limitations, it clearly shows that temperatures at 50 m depth are generally below 0 C on the eastern Barents Sea margin, and >2 C or even >4 C along the western Spitsbergen where the effects of Atlantic Water transport by the West Spitsbergen Current are more directly felt.

A new paragraph early in the manuscript (L65-L7) now describes the oceanographic context in more detail.

R2.7

Line 97. Is it understood that these glaciers calved at the grounding line and do not have any floating section? If so, how do you know and can you please clarify?

This should indeed be clarified. We have added the following sentence to the new paragraph that contains more regional glacier-ocean context for the study: “*No persistent ice shelves or floating tongues have been observed on Svalbard (Dowdeswell et al., 2020) and glaciers on Austfonna are far from the flotation limit when comparing terminus freeboard with bathymetry (e.g. Fig. 1f).*” (L62-L64).

R2.8

Line 104. It would be easy from these studies of undercutting to estimate the rate of undercutting from depth and thermal forcing for your glaciers. Then you could compare with ice velocity and with a rate of undercutting calculated in more "normal" ocean conditions (perhaps much older ocean temp data?). This would help build the quantitative nature of the manuscript which is currently more based on temporal correlation than actual numbers.

While we agree with the reviewer that it would be ideal to quantify the relationships more than we currently do, we do not fully agree that undergoing a quantitative analysis of undercutting based on our data would be trivial - or even very meaningful given the number of parameters available to us (see also our responses to 2.21). We have not undertaken any modelling of undercutting, but we have made attempts to strengthen the quantitative nature of the study where possible and acknowledging the limitations of the study elsewhere (see 2.21).

R2.9

Line 105. Unclear but you have not even tried.

The reviewer is indeed correct that we have not tried to separate melting vs calving – this would require data and analysis outside the scope of the study. We have rephrased the sentence in question from “*We are unable to separate between frontal ablation..*” to “*Our approach does not distinguish between frontal ablation..*” in order to make it clear that we have not attempted to make such a separation (L152).

R2.10

Line 116. Typo, there is no Fig. 2d.

Fixed. Thanks! (L166).

R2.11

Line 120. This could come earlier on. In Greenland, the warm water is found deeper. 100 m is shallow, but you have data that show it. The water that matters is at the grounding line depth of the glaciers. I am not sure that we need to know the temperature at 20 m depth other than to provide contextual information.

Thank you again for the insightful comments.

To the first point, information about the hydrography has been moved earlier (L65-7 from L170-171) and expanded into a wider context in the new paragraph about the glacier-ocean context of Svalbard (see response R2.1). After reviewing the relevant literature and rephrasing the, we have modified the statement about “core depth” (which now specifically refers to the northern Barents Sea) to say 150-200 m rather than 100-150 m.

To the second point, we have three comments:

- 1) We agree that glacier melt rates should be expected to be (much) more responsive to ocean temperatures near the grounding line than at shallower depths. However, this does not mean that temperatures in the ambient water column above the grounding line depth do not matter. Considering buoyant plume theory (Jenkins, 2011), a meltwater plume rising along a glacier front will entrain ambient water along its vertical extent, with entrainment of warmer water resulting in increased melt above the entrainment depth. The effect of warm ambient ocean waters on integrated glacier melt will thus absolutely be greatest near the grounding line, but it is not zero elsewhere.
- 2) We do not actually have measurements of the ambient water column at the glacier face; our measurements represent the offshore coastal ocean more broadly, and isotherms may change depths between the mooring location and the coast. We have added a sentence to this point at L198-199, and additional discussion about the significance of our ocean temperature records follows in L199-203.
- 3) Lastly, it is true that depth-intensified melting may have amplified effects on frontal ablation as a result of undercutting. However, even melt occurring at the ocean surface could be expected to contribute to undercutting since it would be occurring below the above-water freeboard, which for Austfonna can be as high as 50 m (e.g., Fig 1f).

R2.12

Line 139. Again, do you have proper knowledge of ALL the troughs?

In the revised manuscript, we have added much more detail about which areas have poor data coverage for bathymetry, and what the implications are (see our response to point 2.2 above). We have also rephrased the sentence in question changing the wording from “deep (>100 m) troughs” to “a deep (100> m) through” in order to avoid confusion (L189).

R2.13

Line 158. Deep grounding line means more thermal forcing and more access of warm waters (> 100 m) to the glacier base. The authors seem to think that areal melt matters, but that's not true. The melting that matters is the melting near the base, because it will affect basal friction. Melting in intermediate layers do nothing much to the glacier force balance, unless you can explain that to the reader.

We do not entirely follow the reviewer in that areal melt does not matter. In glaciers where terminus melt is important, the submarine melt itself is certainly large enough to be a major factor in the glacier mass balance. Melting in intermediate layers may still cause undercutting (see our response to 2.11 above). And while the glacier force balance near the terminus may play into the process in several ways, it is not necessary to consider glacier dynamics in order to explain frontal ablation driven by ocean thermal forcing.

We do follow the reviewer entirely in that water temperatures near the grounding line are much more important to frontal ablation than e.g. warming near the surface. In line with this, we have modified the sentence in question to: “... *means that warm water at depth can access the glacier base.*” (L220-221).

R2.14

Line 161. You may be right but I don't what this is based on. Speculative? A fast front calves more because .. it has to.

Thank you, we largely agree that this was perhaps unfortunately phrased. We have reformulated the text and included a reference as basis for the attribution to crevassing: “*A fast-flowing front is likely to have higher strain rates, forming fractures and crevasses that can propagate into calving ice blocks, especially in combination with undercutting from submarine melting (e.g., Benn et al., 2007).*” (Paragraph from L219).

R2.15

Figure 4. So these are the big guns that are marine based. They only show up in Figure 4. Could you mention that earlier?

Thank you. The updated Fig 1 now shows all Svalbard calving fronts (panel c). We now also describe the glaciological setting at Austfonna (including the long eastern calving front) in more detail in the new section early on in the manuscript (L59-64).

R2.16

Line 171. Did you document surface runoff vs basal melting of grounded ice from geothermal and friction? It would be good to see these numbers if you have them. Not too hard to get, but no numbers in the paper.

We have no data on basal conditions in this study and due to the speculative nature of this paragraph as you point out below, we have removed it.

R2.17

I found the whole paragraph rather speculative.

We agree with the reviewer upon reevaluating this paragraph. It has been cut in its entirety from the revised manuscript (L236-L241).

R2.18

Line 178. Sea ice is important as it modulates the production of salty, cold water which might modulate the incursions of Atlantic water to the glaciers. This "making" effect is not mentioned, but it might be of interest to quantify the sea ice cover in front (to be defined) of these glaciers. Numerous papers have shown this to be an important factor. Could you at least mention that?

To the reviewer's point, we have added the following sentence to the revised manuscript in a section discussing coastal ocean circulation toward the glacier fronts (L197-198): *"Wind, sea ice melt and formation, and tidal forcing could conceivably all play a role in mediating the nearshore ocean circulation"*.

We have also included a correlation between FA and sea ice concentration (Fig.S3) and an accompanying sentence describing it (L101-104).

A time series of sea ice concentration, as well as a map of mean sea ice concentration in the area offshore of eastern Austfonna is shown in Supplementary Fig. S1. In the revised submission, we have also added a map of mean fast ice occurrence from remote sensing. The map (Figure S1c) shows that there is very little fast ice in front of the eastern margin of Austfonna including Storisstraumen.

R2.19

Line 189. But is not it true that this is the only part of Svalbard with marine terminating glaciers? In that case, not a big surprise if the west does not follow the ocean ..

Marine-terminating glaciers are not only found in the east; in fact, most previous studies of Svalbard tidewater glaciers are from the more accessible western side of the archipelago. Thank you for reminding us to clarify this to the reader – we have done so by adding a figure showing all the marine-terminating glacier fronts in Svalbard (Fig 1c) and new glaciological context on line L59-L64.

R2.20

Line 196. Please add references that show that ocean forcing is not included in the projections of Svalbard glacier evolution. I suspect it is true, but quoting a reference would make this statement stronger.

Good point. The sentence has been rewritten to include a reference on this: “*Given that ocean forcing is not included in current region-scale glacier models (Rounce et al., 2023), this development may cause higher than projected mass loss and sea level contribution from high Arctic glaciers.*” (L292-294).

R2.21

Overall this is a great study. I would encourage the authors to make one more step - to the extent possible - to be more quantitative with melt numbers, warm water numbers (e.g. any evidence for long term warming on the east side?), estimates based on simulations would be better than nothing. So many papers make the claim that the ocean matters, they have the ocean forcing, the end result (retreat), but do not try to connect the numbers. Why? If it is hard to do, say so, and mention that loud and clear, and then make sure to collect more data (e.g. on melt rates of grounded ice) in the future; otherwise accept - and state - that the inference is only a temporal correlation at this stage. That's important to state.

Thank you very much for these useful and encouraging comments. We have taken several steps to strengthen the quantitative parts of the study (detailed below). However, the reviewer is ultimately correct in that the main line of evidence here relies on the temporal character of the observational time series during the study period. We have tried to make it explicit that that is the case.

We have made the temporal connections we describe more quantitatively substantiated by including correlations and scatter plots comparing FA at Storisstraumen with environmental parameters including ocean temperatures and runoff (Fig S3, L99-104).

In the revised manuscript, we have included frontal ablation time series for all of Austfonna’s marine-terminating glaciers, allowing us to compare their relative magnitudes and temporal characteristics (L210-218, Fig 4c). In general, quantitative comparisons have been made where we believe they are appropriate given the available data (e.g. comparing the relative contributions of terminus mass loss and freshwater runoff and comparing magnitudes of frontal ablation across Austfonna glaciers), but we have attempted not give the impression that we have information beyond what we believe are the limitations of the available data, in line with the reviewer’s comment.

Meaningful simulations to pin down the quantitative relationships would in our view both require a technical effort well beyond the scope here (e.g., developing a realistic coupled ocean-glacier model spanning coastal Austfonna), and additional supporting observations (e.g., obtaining realistic boundary conditions for a large number of ocean and glacier parameters). Of course, we hope that this study can motivate future studies both in the observational and modelling directions.

We completely share the reviewer’s view that limitations should be clearly acknowledged and that it should be fully explicit how we arrive at our conclusions. We have made several changes to the wording throughout the manuscript to this end. In a new

paragraph (L262-269) where we to put our study in a broader context, we now state explicitly that "*..the data and methods presented here focus on broad seasonal covariances rather than quantitative relationships, [but] they do provide qualitative insights into relevant to understanding glacier-ocean interactions more broadly*". Directly below, we also summarize the most important conclusions we believe we can reasonably draw from our data ("*Our findings indicate that i) ocean forcing is the primary driver of frontal ablation at Austfonna, and ii) subglacial runoff has limited impact on frontal ablation when the thermal driving is very small*"). Further modifications throughout, including at the start of the manuscript (L55-57, L75-77) hopefully make it clearer to the reader exactly what we are attempting to do.

As for evidence of long-term warming, we find it necessary to lean on Lind et al., 2018, who used hydrographic data going back to the 1970s to show long-term ocean warming, which intensified in the mid-2000s. A similar conclusion is drawn in Ingvaldsen et al., 2021. Both references are included early in our manuscript (L30).

Reviewer 3

Dear Manuscript Authors and Nature Communications Editors,

In this study, Foss et al investigate drivers of ice front variability at glaciers draining the Austfonna ice cap, finding that oceanic (rather than atmospheric) temperature variability is responsible for modulating glacier ablation. The authors leverage a unique mooring dataset spanning a few years in addition to remotely-sensed data of advance/retreat and ice velocity. While many of the glaciers on the ice cap are considered, the main focus and majority of the study discusses Storisstraumen glacier.

In reading this manuscript, I found the study to be a robust interpretation of the data and an interesting addition to a broader narrative of studies linking ice loss and sea level rise to the “Atlantification” of the Arctic. The “story” is clear, as are the figures and analysis of data.

However, the scope of the study is quite limited – the study is focused on a handful of glaciers (mostly one) on a single ice cap in the Arctic for a timespan of a few years. The temporal and spatial scope is largely based on the time duration and location of the available mooring data. I acknowledge that maintaining (and funding!) a mooring in this location is quite a difficult pursuit but it should be noted that the mooring data is not novel to this study as the lead author has already published a separate manuscript on this data (Lundesgaard et al 2022).

In considering this paper for Nature Communications, I am not sure the scope is sufficient as it’s written. Other papers on the same topic published in this journal tend to be extended in time (e.g. Millan et al 2023) and/or extended in space to be pan-Arctic (e.g. Kochitzky et al 2023). Given this precedent, I would recommend this study for publication in this journal if the observational results were leveraged to analyze a longer time period (decades) and/or potentially more glaciers (at least all of Svalbard/Austfonna). There are decades of available remotely sensing data which could be used to quantify glacier evolution. Similarly, a comparison of the mooring data with available ocean state estimates (e.g. from NEMO, ECCO, etc) would likely allow an extension of the analysis over similar periods and large spatial areas using model output. These two records would provide a valuable extension and would likely raise the impact of this study significantly.

Overall, I would recommend this paper for publication after the major revisions described above. If that path is taken, I would be more than happy to review a revised version of this study.

Sincerely, Mike Wood

Thank you for your comments - we appreciate the time and effort you have invested in reviewing our work.

Following your suggestion, we conducted a comparison between the mooring data and available ocean reanalysis products, detailed in Appendix I below. The conclusion from

this brief analysis is that while the reanalysis products may provide a decent representation of upper ocean temperatures (which are largely controlled by surface fluxes and sea ice processes), it is not feasible to use them at greater depth, where temperatures are influenced by horizontal advection and vertical mixing processes. Reanalysis temperatures near 100 m depth are generally significantly warmer than observations through most of the year, and particularly fail to reproduce the near-freezing temperatures observed in the mooring data during spring. Since melt rates can be expected to be proportional to the thermal driving, the water temperature above freezing, we unfortunately do not consider it feasible to use ocean reanalysis products to examine seasonal ocean-glacier forcing in this area.

A more comprehensive analysis would be necessary in order to pinpoint why these models fail to reproduce a realistic hydrography in the northwestern Barents Sea. We note that the region is challenging to model due to the inherently small scales, the relative scarcity of ocean observations in winter, the complexities of AW transport along narrow topographically steered currents affected by instabilities and wind forcing, and sea ice interacting dynamically and thermodynamically with the atmosphere and ocean. In particular, the sub-mixed-layer hydrography is strongly influenced by advection of Atlantic Water. Models in general struggle to reproduce the Atlantic Water layer in the Arctic Ocean (e.g., Dörr et al., 2024; Ilıcak et al., 2016; Muilwijk et al., 2023), and it is perhaps not a surprise that this is also the case in our study area, where the source of Atlantic Water is likely the current branch which passes through the Fram Strait and travels along the continental margin of the Nansen Basin in the Arctic Ocean.

Year-round radar imagery for glacier observations is only freely available after the launches of Sentinel-1A and 1B in April 2014 and 2016, respectively. While front positions could be obtained for earlier periods using other data sources, these would not enable us to examine the evolution through the season as we are able to do in the Sentinel era. Moreover, since it has not proved feasible to use ocean reanalysis products or the analysis, our comparison between ocean temperatures and glacier behaviour will necessarily be limited to the time span of the mooring (from 2018).

For the reasons above, we have not extended our study backwards in time. However, we have extended the calculation of frontal ablation to the entire ice cap as suggested by the reviewer. From this, we get a 5-year time series of frontal ablation for all marine-terminating glaciers in Austfonna, spanning a variety of environmental settings (Fig 4c and associated discussion several times in the revised text, notably at L209-217). This constitutes the first records of frontal ablation for these glaciers that resolve seasonal variations.

Additionally, we have significantly revised the manuscript to place our study within a broader context, including comparisons with other research and a discussion of the implications of our findings in relation to plume theory (esp L262-282; see also the responses to Reviewers 1 and 2). We have also introduced a new section describing the unique physical environment of our study area.

Thank you once again for your valuable feedback, which has contributed to improving the manuscript. We hope these revisions address your concerns, and look forward to your continued review.

Appendix I: Comparison with ocean reanalysis products

We investigate whether ocean reanalysis products can be used to extend the study backward in time beyond the start of the mooring record in 2018. We compare ocean temperature from the models with observed in-situ temperatures at the M1 mooring in order to get an impression of how well the models represent the seasonal ocean temperature variations.

After looking into the various available reanalysis products, we decided to investigate two ocean reanalysis products, the *TOPAZ4b Arctic Ocean Physics Reanalysis* and the *MERCATOR GLORYS12V1 Copernicus Marine Global Ocean Physics Multi Year Product*. Both are relatively high resolution, overlap with the study period of 2018-2022 (only until 2021-06-30 in for the latter product) and extend backwards to the early nineties, and both are commonly used in the Arctic Ocean. (The most recent version of ECCO state estimate ends on 2018-01-01 and does not overlap with the mooring data - we have therefore not included it here).

Reanalysis products

TOPAZ Arctic Ocean Physics Reanalysis

We examine the daily [TOPAZ Arctic Multiyear Physics product](https://doi.org/10.48670/moi-00007) (<https://doi.org/10.48670/moi-00007>), which is based on the HYCOM ocean model coupled to a single-layer sea ice model. The model covers the North Atlantic and Arctic Oceans with output on a 12.5 km horizontal grid with 40 vertical levels and covers the time period from 1991 through 2022. The model assimilates data from various sources including sea surface height and temperature, in-situ ocean profiles from cruises and moorings, and sea ice concentration, thickness, and drift from remote sensing.

We use the TOPAZ4b product available from the Copernicus Marine Service. Data are accessed using the Copernicus Marine Toolbox Python API.

Mercator / Copernicus Marine global ocean reanalysis

The operational Copernicus Marine global ocean reanalysis product (<https://doi.org/10.48670/moi-00021>) provides daily output at 1/12° horizontal resolution and 50 fixed depth levels from 1993 to 2021-06-30¹ The operational Copernicus Marine global ocean reanalysis product is global but has been used in several studies examining the western Nansen Basin including the boundary current north of Svalbard (Athanasé et al., 2020; Bertosio et al., 2020; Koenig et al., 2017). The system is based on a NEMO3.1 ocean model and LIM2 sea ice model. Atmospheric forcing is from ERA-Interim and ERA-5. Various observations are assimilated, including sea surface height and temperature, in-situ ocean profiles, and sea ice concentration from remote sensing.

¹ This applies to the reanalysis product, *GLOBAL_MULTIYEAR_PHY_001_030*. Operational/interim products extend nearer to real-time.

We look at a current version of the reanalysis output, the GLORYS12V1 product. Data are accessed using the Copernicus Marine Toolbox Python API.

Comparison with temperature at the M1 mooring

We compare the output from each model with the ~20 m and ~95 m temperature time series from the M1 mooring. We extract the model time series from the horizontal grid cell centered nearest to the M1 location, and the vertical cell nearest to 20 and 95 m, respectively².

The comparison is done in terms of potential temperature, which is the variable output by both models. Conversion of mooring data from in-situ temperature to potential temperature was done using the GSW-python function `pt_from_t()`.

Figure A1 shows a comparison of ocean potential temperature at ~20 and ~95 m depth. The reanalyses reproduce the observations reasonably well in the upper ocean, although there are notable discrepancies in amplitude between models and observations as well as between Mercator and Topaz. At ~95 m depth, both reanalysis products reproduce a seasonal cycle with maximum amplitude in late autumn to winter, but the comparison between models and against observed temperatures is otherwise poor. In particular, the spring baseline of near-freezing temperature in observations is not replicated in the reanalysis products.

² The nearest Mercator grid cell is centered 2 km from M1, and we examine the time series from 18.5 and 92.3 m, respectively. The nearest TOPAZ 4b grid cell is located 4 km from M1, and we examine the depth levels 20 and 100 m.

Figure A1: Comparison of potential temperature observed at the M1 mooring and from two ocean reanalysis products. Reanalysis temperatures are extracted from the grid cell centered nearest to the location of M1, and at the depth level centered closest to 20 and 95 m, respectively (the exact model depth levels are shown in the figure legends).

Appendix II: Retreat rates with independent-y scales

Figure A2: Retreat rates for all glaciers (shown in Figure 4b) with y-axis scales normalized to the retreat/advance range of each glacier.

References

- Athanase, M., Provost, C., Pérez-Hernández, M. D., Sennéchaël, N., Bertosio, C., Artana, C., Garric, G., & Lellouche, J.-M. (2020). Atlantic water modification north of Svalbard in the Mercator physical system from 2007 to 2020. *Journal of Geophysical Research: Oceans*, *125*(10), e2020JC016463.
- Benn, D. I., Warren, C. R., & Mottram, R. H. (2007). Calving processes and the dynamics of calving glaciers. *Earth-Science Reviews*, *82*(3), 143–179. <https://doi.org/10.1016/j.earscirev.2007.02.002>
- Bertosio, C., Provost, C., Sennéchaël, N., Artana, C., Athanase, M., Boles, E., Lellouche, J.-M., & Garric, G. (2020). The Western Eurasian Basin Halocline in 2017: Insights From Autonomous NO Measurements and the Mercator Physical System. *Journal of Geophysical Research: Oceans*, *125*(7), e2020JC016204. <https://doi.org/10.1029/2020JC016204>
- Boyer, T., Baranova, O., Biddle, M., Johnson, D., Mishonov, A., Paver, C., Seidov, D., & Zweng, M. (2015). Arctic Ocean regional climatology (NCEI Accession 0115771). *NOAA National Centers for Environmental Information (NOAA, Washington, DC)*.
- Dörr, J., Årthun, M., Eldevik, T., & Sandø, A. B. (2024). Expanding Influence of Atlantic and Pacific Ocean Heat Transport on Winter Sea-Ice Variability in a Warming Arctic. *Journal of Geophysical Research: Oceans*, *129*(2), e2023JC019900. <https://doi.org/10.1029/2023JC019900>
- Dowdeswell, J. A., Ottesen, D., & Bellec, V. K. (2020). The changing extent of marine-terminating glaciers and ice caps in northeastern Svalbard since the ‘Little Ice Age’ from marine-geophysical records. *The Holocene*, *30*(3), 389–401. <https://doi.org/10.1177/0959683619887429>
- Dunse, T., Schellenberger, T., Hagen, J. O., Kääh, A., Schuler, T. V., & Reijmer, C. H. (2015). Glacier-surge mechanisms promoted by a hydro-thermodynamic feedback to summer melt. *The Cryosphere*, *9*(1), 197–215. <https://doi.org/10.5194/tc-9-197-2015>
- Ilicak, M., Drange, H., Wang, Q., Gerdes, R., Aksenov, Y., Bailey, D., Bentsen, M., Biastoch, A., Bozec, A., Böning, C., Cassou, C., Chassignet, E., Coward, A. C., Curry, B., Danabasoglu, G., Danilov, S., Fernandez, E., Fogli, P. G., Fujii, Y., ... Yeager, S. G. (2016). An assessment of the Arctic Ocean in a suite of interannual CORE-II simulations. Part III: Hydrography and fluxes. *Ocean Modelling*, *100*, 141–161. <https://doi.org/10.1016/j.ocemod.2016.02.004>
- Ingvaldsen, R. B., Assmann, K. M., Primicerio, R., Fossheim, M., Polyakov, I. V., & Dolgov, A. V. (2021). Physical manifestations and ecological implications of Arctic Atlantification. *Nature Reviews Earth & Environment*, 1–16. <https://doi.org/10.1038/s43017-021-00228-x>

Jackson, R. H., Motyka, R. J., Amundson, J. M., Abib, N., Sutherland, D. A., Nash, J. D., & Kienholz, C. (2022). The Relationship Between Submarine Melt and Subglacial Discharge From Observations at a Tidewater Glacier. *Journal of Geophysical Research: Oceans*, *127*(10), e2021JC018204.

<https://doi.org/10.1029/2021JC018204>

Jenkins, A. (2011). Convection-Driven Melting near the Grounding Lines of Ice Shelves and Tidewater Glaciers.

Journal of Physical Oceanography, *41*(12), 2279–2294. <https://doi.org/10.1175/JPO-D-11-03.1>

Koenig, Z., Provost, C., Sennéchaël, N., Garric, G., & Gascard, J.-C. (2017). The Yermak Pass Branch: A Major

Pathway for the Atlantic Water North of Svalbard? *Journal of Geophysical Research: Oceans*, *122*(12),

9332–9349. <https://doi.org/10.1002/2017JC013271>

Lind, S., Ingvaldsen, R. B., & Furevik, T. (2018). Arctic warming hotspot in the northern Barents Sea linked to

declining sea-ice import. *Nature Climate Change*, *8*(7), 634–639. [https://doi.org/10.1038/s41558-018-](https://doi.org/10.1038/s41558-018-0205-y)

[0205-y](https://doi.org/10.1038/s41558-018-0205-y)

Muilwijk, M., Nummelin, A., Heuzé, C., Polyakov, I. V., Zanowski, H., & Smedsrud, L. H. (2023). Divergence in

Climate Model Projections of Future Arctic Atlantification. *Journal of Climate*, *36*(6), 1727–1748.

<https://doi.org/10.1175/JCLI-D-22-0349.1>

RESPONSE TO REVIEWERS, NATURE COMMUNICATIONS

This document contains the response to the second round of reviews received for our submitted manuscript *Ocean warming drives immediate mass loss from calving glaciers in the high Arctic*. The original comments from the three reviewers are shown in black, with our numbering for easy reference. Author responses are shown in blue.

Line numbers refer to the track-change PDF document.

Reviewer 1:

Thanks to the authors for addressing the points raised in my earlier review. I think that these have strengthened the manuscript, particularly through some considered discussion of the new findings in the context of existing studies. I had only a few specific comments on the earlier manuscript and these have to a large extent been addressed effectively. I have just a few minor points of clarity on the current version.

Thank you very much for the constructive review and the useful comments! Responses to the individual points are found below.

R1.1

L78-79. As I think I flagged in my previous review, the early part of the manuscript lacks explanation of the mechanisms that might be expected to affect frontal ablation. As such, if you're not already versed in the subject, it's not obvious at this point why one might expect meltwater runoff to be a major control on frontal ablation. Meltwater plumes are mentioned later (e.g. L140), but I think it would be helpful to have a very brief explanation earlier in the paper of why meltwater discharge has been hypothesized as a key control, given it is a key focus on the study.

Thank you for the suggestion. In the revised manuscript, we have added some more background content in the beginning of the manuscript (L48-L55) in order to better explain why we focus on freshwater runoff and ocean temperatures as potential drivers:

“Seasonal variations in frontal ablation rates typically result from a combination of changes in the ocean water column at the glacier front and subglacial freshwater runoff, both of which exhibit strong seasonality (e.g. Jackson 2022, Greene 2024). Frontal ablation is the sum of calving and melting, the latter being strongly influenced by buoyant meltwater plumes that rise along the glacier terminus, drawing in ambient ocean water and transferring heat from the plume to the ice (Straneo and Cenedese 2015, Truffer and Motyka 2016, Hewitt 2020). Consequently, frontal melt generally increases with both ocean temperature and freshwater flux. The relative importance of these two driving factors—one atmospherically driven and the other oceanic—can vary significantly across different systems.”

R1.2

L200. Thank you for providing the additional figure on this point. I agree that the western glaciers show little obvious seasonality, but I do think they show some synchronicity with each other and the other sectors with respect to periods of major retreat or advance (e.g. the major retreat phase during mid 2018). I'm not going to demand that you include this point, but I do think the data from these fjords supports your broader hypotheses more than you allow for in your description.

Thank you for your insightful comment. We agree that the western glaciers show some synchronicity with other sectors, particularly during the 2018 retreat phase. While it would be possible to elaborate on this point, we have maintained the current description for conciseness, as additional detail here would not directly contribute to elaborating the paper's central argument.

R1.3

Figure S3. It's striking from this figure and the computed correlations that the correlation between frontal ablation and ocean temperature is much stronger at 20 m than at 95 m (which is no better than air temperature as a predictor). I think this merits a short discussion, not least to counter any criticism that the conclusions are overly sensitive to which depth you happen to select for water temperature. Do you think there is a physical explanation for this, or is it due to data quality, or something else?

We think this is likely a result of lateral differences between the offshore mooring location and the coastal water column that is actually exposed to the glacier face. We comment on the difference in correlation and potential reasons in a paragraph later on in the manuscript (L187-L195). However, we agree that it is also worth a mention when the correlations are presented and have added a paragraph at L105-109;

“The poorer correlation at 95 m depth may be a result of lateral gradients between the offshore temperatures and the water column near the glacier face. Although we will show below that deep heat transport toward the coast does occur, we do not know whether it occurs systematically or throughout all seasons, and the mixed layer depth may for example deepen toward the coast during parts of the year.”

R1.4

Figure S1a. Having noted the correlation between air temperature and frontal ablation in Figure S3, it would be useful if this was plotted alongside the other main timeseries in Figure 2 (or if these were added to Figure S1). It's not possible to meaningfully compare the timing of seasonal trends between the two figures as it stands

Thank you for the suggestion. We have added air temperature from Kvitøya weather station as a new panel d) in Figure 2.

Reviewer 2

I am satisfied with the revision, except for one point that I wanted to clarify again. Apart from this point, which the authors are welcome to take into account or not, I am ok with the publication of the paper. I do not need to see their response.

The authors state "We do not entirely follow the reviewer in that areal melt does not matter. In glaciers where terminus melt is important, the submarine melt itself is certainly large enough to be a major factor in the glacier mass balance. Melting in intermediate layers may still cause undercutting (see our response to 2.11 above). And while the glacier force balance near the terminus may play into the process in several ways, it is not necessary to consider glacier dynamics in order to explain frontal ablation driven by ocean thermal forcing."

In terms of ice melt, yes, frontal melting matters, but as long as the ice crosses the (fictitious) grounding line, it does not matter if ice melts or breaks up in the ocean, it will contribute to sea level rise. More important is the effect of melt on the glacier stability, because this will determine if the glacier will speed up or not in response to an increase in melt. Frontal melt distributed along the ice face does not matter in the force balance, while melting at the grounding line matters greatly. If you remove grounded ice, you reduce basal friction, which effects the glacier force balance. Conversely, melt at the water line does not matter, it leaves the majority of the ice column below water friction at the bed. The ice above it, that's typically about 10% of the ice thickness. If you melt ice right at the grounding line, however, the entire column of ice above the cut will no longer affect basal friction, which is the intrinsic power of undercutting

I do not know if I explained this well enough, but I hope the authors will give it some thought. It is important to support the point that the shallow water temperature do not matter; the one that matters is the water temperature at the depth of the grounding line.

A very good explanation, thank you for clarifying. We do take your point that in terms of the long-term stability of the glaciers, melting near the grounding line largely is what matters, and agree that this should be stated in the manuscript.

In the revised version of the manuscript, we have added a sentence to this effect (L249-251): *"In particular, ocean-driven melting that occurs near the grounding line may affect basal friction and the force balance near the terminus, which in turn may result in acceleration of the glacier and cause dynamic thinning of upstream areas."*

Reviewer 3

Dear Manuscript Authors and Nature Communications Editors

The revised manuscript of “Ocean warming drives immediate mass loss from calving glaciers in the high Arctic” by Føss et al has undergone significant improvements since the initial draft. Here, I will briefly summarize my initial comments and my impressions on the responses from the authors.

My primary comment on the initial draft was that the paper was limited in scope relative to other papers submitted to Nature Communications. This viewpoint was echoed in some of the comments provided by the other reviewers on the manuscript. To expand the scope, I suggested that the study could be extended to all glaciers in Svalbard, and that the authors could leverage historical imagery and ocean reanalysis products to extend the study backwards in time.

For the number of glaciers in Svalbard, the authors have expanded their study to all of the glaciers, now providing additional timeseries in the revision. This has provided a significant improvement to the scope of the study.

For the extension backwards in time, the authors conducted a comparison between the mooring and reanalysis products in the region and have even provided some figures for the comparison. The authors have concluded that the reanalysis products are not trustworthy at depth (100m) and therefore that the extension backward in time is not possible. In looking at the timeseries however, the reanalysis products do seem to be able to capture interannual variability at depth even though they do not get the absolute magnitude quite right. I would imagine that if the timeseries were extended backward in time, the “Atlantification” mentioned in this paper would be visible in both temperature and salinity. Further, since the paper has already linked ocean temperature with the glacier retreat, I would suspect the glaciers would retreat in concert with the warming trend, and that finding would significantly bolster the impact of the study. However, I acknowledge that this would entail a significant amount of additional work, primarily in the extension of the frontal ablation timeseries.

Given these considerations, I will defer to the editor and other reviewers for the decision for publication. I think the tools are in place to generalize from mooring to ocean reanalysis and this would provide valuable information over several decades during a time of widespread change in the Arctic. However, I also acknowledge that this would require additional work and may be addressed in an additional separate study.

Sincerely,
Mike Wood

Thank you for your thoughtful feedback. We appreciate your acknowledgment of the broadened scope with the extended frontal ablation analysis that we have carried out. We acknowledge that this study does not itself resolve “Atlantification” - it merely documents a responsiveness to ocean thermal forcing, and posits that regional Atlantification therefore can be expected to affect frontal ablation rates at Austfonna and elsewhere in the Eurasian Arctic.

We fully share your view that an analysis of frontal ablation and ocean temperature extending over decades would be extremely valuable. We maintain that the applicability of available reanalysis products is a major limitation: while they do show interannual variability, we are concerned about the accuracy and resolution required to confidently link this variability to glacier frontal ablation. Another limitation is that year-round radar-based frontal ablation estimates that capture the seasonality of glacier change (our primary focus), are only feasible to derive from 2014 (Sentinel-1A) / 2016 (Sentinel-1B).

That said, there are promising developments underway. Higher-resolution ocean models are currently being developed at NPI and elsewhere, and we are optimistic that these may eventually provide observationally validated output that could be integrated with glacier time series further back in time, as well as for projections into the future. We hope that future studies can build upon the foundations laid here and provide the long-term perspective that you rightfully point out would be so valuable.